# Central medial thalamic nucleus dynamically participates in acute itch sensation and chronic itch-induced anxiety-like behavior in male mice

Jia-Ni Li[1,6], Xue-Mei Wu[1,2,6], Liu-Jie Zhao[1,3], Han-Xue Sun[1,4], Jie Hong[1,5], Feng-Ling Wu[1,3], Si-Hai Chen[1,3], Tao Chen[1], Hui Li[1], Yu-Lin Dong ®[1] ✉ & Yun-Qing Li ®[1,2,3,4,5] ✉

Itch is an annoying sensation consisting of both sensory and emotional components. It is known to involve the parabrachial nucleus (PBN), but the following transmission nodes remain elusive. The present study identified that the PBN-central medial thalamic nucleus (CM)-medial prefrontal cortex (mPFC) pathway is essential for itch signal transmission at the supraspinal level in male mice. Chemogenetic inhibition of the CM-mPFC pathway attenuates scratching behavior or chronic itch-related affective responses. CM input to mPFC pyramidal neurons is enhanced in acute and chronic itch models. Specifically chronic itch stimuli also alter mPFC interneuron involvement, resulting in enhanced feedforward inhibition and a distorted excitatory/inhibitory balance in mPFC pyramidal neurons. The present work underscores CM as a transmit node of the itch signal in the thalamus, which is dynamically engaged in both the sensory and affective dimensions of itch with different stimulus salience.

Itch is an annoying sensation that always triggers the urge to scratch, which, like nociception, is composed of sensory, cognitive, motivational, and emotional dimensions[1–4]. Acute itch, which persists for less than 6 weeks, is regarded as a protective mechanism from the surrounding noxious insults (e.g., mosquito bites). Chronic itch can continue for 6 weeks or longer, and induces a series of negative emotions, which severely influence the quality of life in patients[5]. As research on itch develops, a number of key questions remain to be addressed to understand the central itch transmission pathway at the supraspinal level[6]. The parabrachial nucleus (PBN) is an essential relay station that receives approximately 95% of the spinal cord projections[7]. It has been

reported that PBN gets activated by itch stimuli, and that genetic deletion of vesicular glutamate transporter 2 (VGluT2) neurons in PBN impairs scratching behavior in response to both histamine-dependent and histamine-independent pruritogens such as chloroquine (CQ), 5-hydroxytryptamine (5-HT) and bombesin[8]. PBN also transmits pruriceptive information from the dorsal spinal cord to the brain[6]. However, the next transmission node of the PBN concerning itch signals in the forebrain remains unknown.

The central medial thalamic nucleus (CM) is located centrally in the intralaminar thalamus and distinctively populated with large, deeply stained flattened neurons clearly recognizable from the dorsal

[1]Department of Anatomy, Histology and Embryology, K. K. Leung Brain Research Centre, The Fourth Military Medical University, Xi'an 710032, China. [2]Department of Human Anatomy, West China School of Basic Medical Sciences & Forensic Medicine, Sichuan University, Chengdu 610041, China. [3]Department of Anatomy, Basic Medical College, Zhengzhou University, Zhengzhou 450001, China. [4]Department of Human Anatomy, The School of Basic Medical Sciences, Fujian Medical University, Fuzhou 350122, China. [5]Department of Human Anatomy, Baotou Medical College Inner Mongolia University of Science and Technology, Baotou 014040, China. [6]These authors contributed equally: Jia-Ni Li, Xue-Mei Wu. ✉e-mail: donganat@fmmu.edu.cn; deptanat@fmmu.edu.cn

and ventral nuclei[9]. As a pivotal component of the dorsal midline and intralaminar thalamic complex (dMITC), CM has been regarded as a member of the 'limbic thalamus' and has been proven to encode both nociceptive signals and chronic pain-induced negative emotions[10,11]. CM belongs to the medial pain system, and in contrast to the lateral pain system, transmits pain-related noxious feelings[12]. In pruriceptive sensation, despite some differences in activated brain regions, most previous macroscopic imaging and positron emission tomography (PET) results have shown that the thalamus, including the CM, is activated by both acute and chronic itch stimuli[13]. However, limited by temporal and spatial resolution, the participation of CM in itch sensation remains unclear.

The present work aimed to reveal the transmission of itch signals (both sensory and affective) through a brainstem-thalamus-cortex pathway by using complementary neuroanatomical and functional approaches. Our morphological evidence suggests that PBN-CM sends projections to the medial prefrontal cortex (mPFC), a key nucleus in thalamocortical relays[14], and modulates negative emotions[15]. As anxiety-like behavior is one of the symptoms of chronic itch[16], we also attempted to determine the modulatory role of the mPFC, to possibly unveil the underlying mechanism of chronic itch-induced anxiety-like behavior.

## Results

### CM participates in the itch sensation

To verify the participation of CM in pruriception, FOS-immunoreactive (ir) neurons in the rostral-caudal axis of the CM were examined. Histamine (His) and chloroquine (CQ) were used to establish acute histamine-dependent and histamine-independent itch models, respectively. For the chronic itch model, 1-fluoro-2,4-dinitrobenzene (DNFB) was applied to the nape of mice with acetone as a control. The DNFB model did not affect the body weight or the locomotor abilities of mice, yet caused obvious skin lesion. The number of FOS+ neurons increased significantly with effective pruriceptive stimulation (supplementary Fig. 1). Since the most substantial increase in scratches was observed on day 14 after DNFB application, this time point was used in all subsequent behavioral assays in the DNFB-induced chronic itch model.

To further identify the neuronal activity of CM with pruriceptive stimulation, a two-color fiber photometry recording system, which allows simultaneous recording of GCaMP6m fluorescent signals at 470 and 410 nm (for isosbestic control signals), was utilized based on a previous report[17,18] (Fig. 1a). An elevated fluorescent signal innervated by a 470 nm laser was detected at the onset of the scratching behavior in both His- and CQ-induced itch models (Fig. 1b–e). When summarizing the calcium transients of different animals, both the heatmap and the area under the curve (AUC) displayed a significant increase of 470 nm-induced signals during scratching behavior when compared with that in the baseline phase (Fig. 1f, g).

Electrolytic lesion of the CM was then performed to test the modulatory effect of CM on itch sensation and chronic itch-related anxiety-like behavior (Fig. 1h, supplementary Fig. 2). Behavioral results showed that the number of scratches decreased significantly in the lesion group with His or CQ injections (Fig. 1i, j). In the DNFB model, both the skin lesion and the scratching behavior were relieved with the CM lesion (supplementary Fig. 2c, Fig. 1k). Furthermore, the results of the open field test (OFT) and elevated plus maze (EPM) tests demonstrated that the center time, OA time and OA distance decreased significantly in the DNFB+Sham group. On the other hand, the lesion of CM exhibited an increasing trend in OA time and OA distance, although they were not significantly different from those in the DNFB model without CM lesions (Fig. 1l–m).

### CM receives pruriceptive afferents from the PBN

PBN has been regarded as a prominent target of spinal projections carrying pruriceptive information[8]. We were then interested in

knowing whether the CM receives afferents from the PBN. The adeno-associated virus (AAV) fused with *c-fos* as a promoter (activity-dependent) and enhanced with enhanced yellow fluorescent protein (AAV$_{2/9}$-cFos-eYFP) was injected into the right PBN to check the eYFP-labeled projecting fibers induced by itch stimulation. The DNFB model was performed to supply the required strong enough intensity of itch stimuli for the full expression of the virus (Fig. 2a). It was revealed that eYFP-labeled neurons were predominantly observed in the lateral PBN in the DNFB group, which is double-labeled with FOS protein and DAPI (Fig. 2b, c). It was noted that the eYFP-labeled fibers and terminals were more abundant in the CM in the chronic itch group (Fig. 2d), suggesting the potential participation of the PBN-CM pathway in itch signal transmission. These results were further confirmed by injecting retrograde tracer Fluoro-gold (FG) into the CM (supplementary Figs. 3a–c). Combined fluorescence in situ hybridization (FISH) and immunofluorescent histochemical staining revealed that VGluT2/FOS/FG triple-labeled neurons were observed in the PBN, predominantly in the lateral PBN. Meanwhile, it was indicated that about 64.35% of the FG-labeled neurons were activated by itch stimuli, and about 82.03% of the itch-activated CM-projecting neurons in the PBN were VGluT2-positive neurons (supplementary Figs. 3d, e), indicative of the glutamatergic projections from the PBN to the CM in itch information transmission.

We then tested postsynaptic neuronal Ca$^{2+}$ activity in response to itch stimulation in the PBN-CM pathway. This pathway was specifically identified by injecting vector-mediated Cre-dependent anterograde trans-synaptic tracing virus into the PBN (AAV$_{2/1}$-hSyn-Cre), and postsynaptic neuronal activity was examined by injecting AAV expressing a double-floxed inverted open reading frame (DIO) and a genetically encoded calcium indicator (AAV$_{2/9}$-hSyn-DIO-GCaMP7s) into the CM. Optic fibers were implanted into the CM to record calcium population signals using a simultaneous two-color activity recording system (Fig. 2e, f). It was observed that GCaMP7s fluorescence was elevated aligned with the onset of scratching behavior in both His- and CQ-injected groups. The AUC also indicated significant increase of 470 nm-induced calcium transients when comparing the scratch and baseline phases (Fig. 2g–l), suggesting the relevance of the PBN-CM pathway in the transmission of pruriceptive information.

To examine whether activation of the PBN-CM pathway is necessary for itch signal transmission, the same anterograde virus with Cre enzyme was injected into the right PBN. The Cre-dependent expression of the Guillardia theta anion channel rhodopsin 1 protein (GtACR1), which exhibits an inhibitory effect with blue laser stimulation, was injected into the CM (Fig. 2m, supplementary Figs. 3f, g). In acute itch models, optogenetic inhibition of the PBN-CM pathway significantly attenuated the scratching behavior induced by both His and CQ (Fig. 2n, o), yet not that in the DNFB group (supplementary Fig. 3h). Specific PBN-CM inhibition also has little effects on the chronic-itch induced anxiety-like behavior (supplementary Figs. 3i, j). These data indicate that the activity of the PBN-CM pathway is necessary for scratching behavior, especially in acute itch models.

### A PBN-CM-medial prefrontal cortex (mPFC) pathway transmits itch signals

To examine the main termination regions for the efferent fibers from the CM neurons receiving PBN projections, the Cre-dependent anterograde trans-synaptic virus was injected into the right PBN, and AAV$_{2/9}$ expressing DIO-eGFP was injected into the CM (Fig. 3a–c). The projecting fibers originating from the postsynaptic neurons in the PBN-CM pathway were then examined. We found that the mPFC is a nucleus with dense projecting fibers, in which the projecting fibers and terminals are concentrated in layers I and II/III (Fig. 3d). The PBN-CM-mPFC pathway was confirmed using retrograde tracing. AAV$_{2/9}$-hSyn-eGFP-Synaptophysin-mRuby (labeling the PBN projecting fibers with eGFP and confirming the synaptic connectivity by mRuby) was injected into

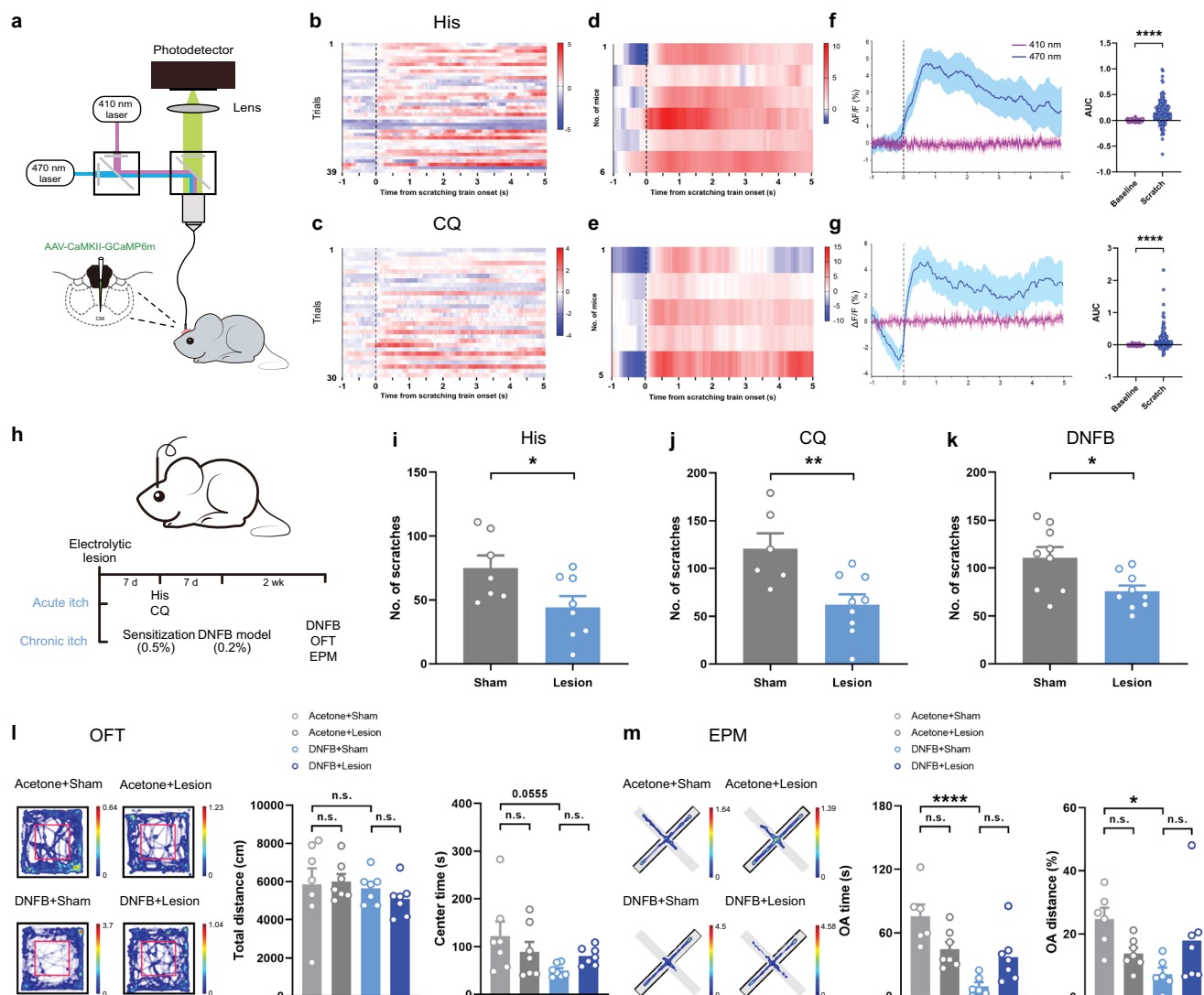

**Fig. 1 | The participation of the CM in pruriceptive sensation. a** Schematic for fiber photometry recording system that allows simultaneous imaging of population calcium signal at 470 nm and 410 nm, with GCaMP6m virus injected into the CM. **b, c** Representative heatmaps showing the calcium response of CM neurons to Histamine (His; $n = 39$ trials from one mouse) and chloroquine (CQ; $n = 30$ trials from one mouse). Color scale at the right indicates ΔF/F (%). **d, e** Heatmaps showing the calcium response in each mouse (His: $n = 6$ mice; CQ: $n = 5$ mice). Color scale at the right indicates ΔF/F (%). **f, g** Quantification of calcium signal (ΔF/F%) for scratching event in His- or CQ-injected itch models (left). Area under curve (AUC; $P < 0.0001$, $n = 169$ trials from 6 mice in **f**; $P < 0.0001$, $n = 161$ trials form 5 mice in **g**) depicting calcium transients of scratching behavior in His and CQ models (right). **h** Representative schematic diagram of electrolytic lesion of the CM, and the timeline concerning behavioral tests. **i, j** Summary data showing changes of the number of scratches in acute itch models induced by His or CQ (Sham vs. Lesion in His group: $P = 0.0353$; $n = 7$, 8 per group; Sham vs. Lesion in CQ group: $P = 0.0072$; $n = 6$, 9 per group). **k** The scratching behavior in 1-fluoro-2,4-dinitrobenzene

(DNFB)-induced chronic itch model ($P = 0.0138$; $n = 9$ per group). **l** Movement tracks and summary data of total distance and center time in open field test (OFT). (Total distance: F(3,24)=0.6264, $P = 0.6050$, Ace (Sham vs. Lesion): $P = 0.997$, DNFB (Sham vs. Lesion): $P = 0.8584$, Sham (Ace vs. DNFB): $P = 0.9918$; Center time: F(3,24) = 2.485, $P = 0.0850$, Ace (Sham vs. Lesion): $P = 0.6121$, DNFB (Sham vs. Lesion): $P = 0.6648$, Sham (Ace vs. DNFB): $P = 0.0555$; $n = 7$ per group). **m** Movement tracks and summary data in elevated plus maze (EPM). (OA time: F(2,16) = 13.95, $P = 0.0003$, Ace (Sham vs. Lesion): $P = 0.1084$, DNFB (Sham vs. Lesion): $P = 0.101$, Sham (Ace vs. DNFB): $P < 0.0001$; OA distance: F(3,22) = 3.331, $P = 0.0382$, Ace (Sham vs. Lesion): $P = 0.3646$, DNFB (Sham vs. Lesion): $P = 0.2267$, Sham (Ace vs. DNFB): $P = 0.0236$; $n = 6$, 7, 6, 7 per group). The color of the pixel in heatmaps in **l** and **m** represents the cumulated time the mice stayed at this position (s). n.s.: no significance, *$P < 0.05$, **$P < 0.01$, ****$P < 0.0001$. Data are presented as mean ± S.E.M. Two-tailed, paired, Student's $t$-test for **f** and **g**, two-tailed, unpaired, Student's $t$-test for **i, j, k**, and one-way ANOVA with Tukey's multiple comparison tests for **l** and **m**.

the PBN, with retrograde tracer FG injected into the mPFC, which is in close contact with FG-labeled neurons in CM (Fig. 3e–g).

The mPFC is involved in the regulation of somatosensory information and affective changes. Therefore, we tested whether activation/inhibition of the CM-mPFC pathway changes itch and itch-related affective behavior. The modulatory role of the CM-mPFC pathway was verified using designer receptor exclusively activated by designer drugs (DREADD) strategy. The vector-mediated Cre-dependent retrograde tracing virus was injected into the right mPFC, and

viruses fused with the genetically engineered muscarinic receptor hM3Dq or hM4Di with DIO were injected into the CM. The effectiveness of the viruses was confirmed using electrophysiological methods with clozapine-N-oxide (CNO) applied in current-clamp recording (Fig. 4a–c). In DIO-hM3Dq-injected mice, the number of scratches remained unchanged in spontaneous, His-, and CQ-injected groups, as well as in the DNFB-induced chronic itch model (Fig. 4d–g). However, chemogenomic silencing of the CM-mPFC pathway dramatically attenuated the scratching behavior in the spontaneous, His- and CQ-

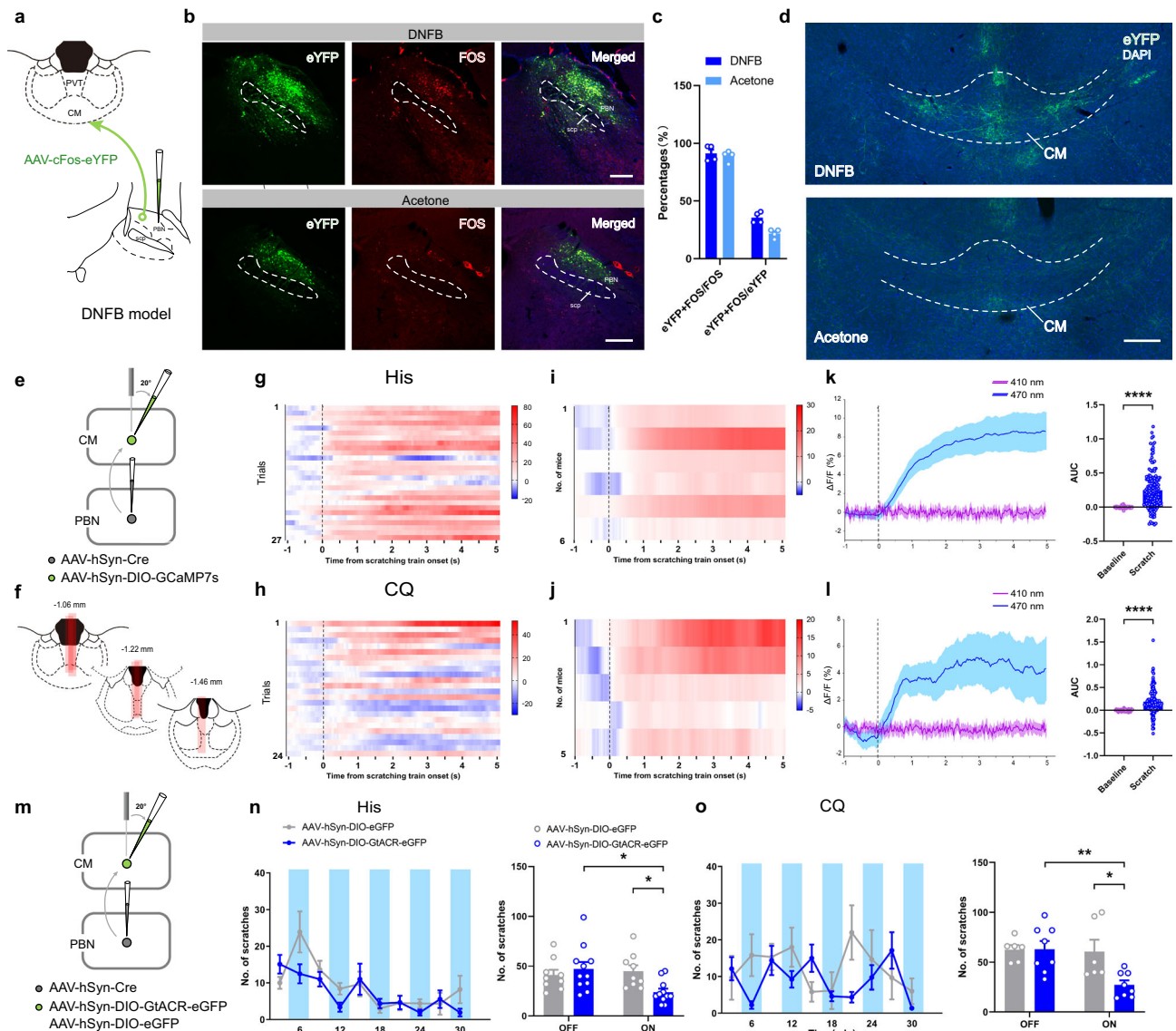

**Fig. 2 | The CM receives itch signal from the PBN. a** Representative schematic diagrams of AAV-cFos-eYFP injection into the PBN of mice performed with DNFB model. **b** Injection sites depicting the location of itch-activated AAV virus (green) and FOS-immunoreactive (red) neurons with DAPI delineating the PBN. Scale bar=200 μm. **c** Percentages of the eYFP/FOS double-labeled neurons that overlap with FOS- or eYFP-expressing neurons in the PBN ($n = 4$). **d** Distribution of eYFP-labeled fibers and terminals in DNFB and Acetone groups ($n = 4$; white dashed lines indicating the CM). Scale bar = 200 μm. **e, f** Schematic diagram and the optic fiber implantation sites of fiber photometry of the PBN-CM pathway. **g, h** Representative heatmaps of calcium response aligned with scratching behavior. Color scale at the right indicates ΔF/F (%). **i, j** Average fluorescent signal in response to His or CQ stimuli in each mouse ($n = 6$, 5 per group). Color scale at the right indicates ΔF/F (%). **k, l** Quantification of calcium signal (ΔF/F %) for scratching event in His- or CQ-injected itch models (left). Area under curve (AUC; $P = 0.0001$, $n = 156$ trials from 6

mice in **k**; $P < 0.0001$, $n = 114$ trials from 5 mice in **l**) depicting calcium transients of scratching behavior with His or CQ stimuli (right). **m** Schematic plot of the virus strategy used for optogenetic silencing of the PBN-CM pathway. **n, o** The changing of scratching behavior in 3 min intervals in light on (blue shaded, 473 nm) and light off periods, with the total number of scratches summarized in light on (15 min) and light off (15 min) phases (His: interaction: $P = 0.0191$, treatment: $P = 0.1834$, light-on (eGFP vs. GtACR): $P = 0.0498$, GtACR (light-off vs. light-on): $P = 0.0166$; $n = 9$, 11 per group; CQ: interaction: $P = 0.0374$, treatment: $P = 0.0429$, light-on (eGFP vs. GtACR): $P = 0.0253$, GtACR (light-off vs. light-on): $P = 0.0083$; $n = 6,8$ per group). $^*P < 0.05$, $^{**}P < 0.01$, $^{****}P < 0.0001$. Data are presented as mean ± S.E.M. Two-tailed, paired, Student's $t$-test for **k** and **l**, and two-way ANOVA with Tukey's multiple comparison tests for **n** and **o**. CM: central medial thalamic nucleus; PBN: para-branchial nucleus; PVT: paraventricular thalamic nucleus; scp: superior cerebellar peduncle.

injected groups, and that in the DNFB model (Fig. 4h–k). Specifically inhibiting the projections from CM to PNs in mPFC also relieved acute itch (supplementary Figs. 4b–d), indicating the necessity of the CM-mPFC pathway in itch sensation. Since the chronic itch model could induce negative emotions, such as anxiety-like behavior, the OFT and EPM were also performed. Although no significant differences were observed in the total distance, the center time in the OFT and the OA time in the EPM increased significantly in DNFB group following CNO administration (Fig. 4l–m). Meanwhile, no significant differences were

observed in time spent in open arm while silencing the CM-mPFC pathway in naïve mice (supplementary Fig. 4a).

## Acute and chronic itch activate the mPFC-projecting CM neurons

To verify the characteristics of mPFC-projecting CM neurons, an electrophysiological technique was used. First, AAV$_{2/9}$ expressing channelrhodopsin (ChR2), a light-sensitive cation channel, was injected into the PBN and retrograde tracer 488-retrobeads were injected

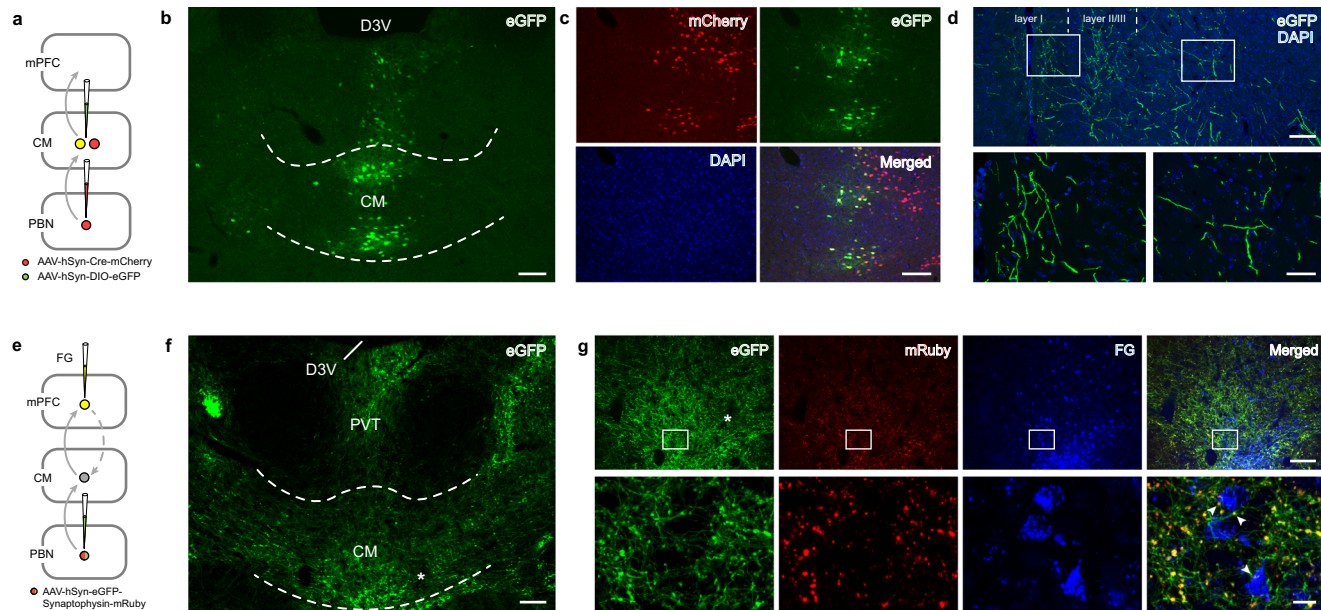

**Fig. 3 | Morphological evidence of the PBN-CM-mPFC neuronal pathway.**
**a** Diagram depicting the virus strategy used to delineate the PBN-CM-mPFC pathway. **b, c** Representative images of virus injection sites, and Cre enzyme (red)/eGFP(green)/DAPI triple-labeled neurons in the CM. Scale bar = 100 µm in **b** and **c**.
**d** Representative images of eGFP-labeled fibers and terminals in medial prelimbic cortex (mPFC). White dashed lines are used to delineate the layer I and layer II/III. The white squares are enlarged as the lower images. Scale bars = 100 µm (upper), 40 µm (lower). **e** Schematic diagram of the PBN-CM-mPFC neuronal pathway by injecting retrograde tracer FG and anterograde virus that labels projecting terminals and synaptophysin simultaneously. **f, g** Representative images of eGFP-labeled projections from the PBN (white asterisk shows the same position in **f** and **g**), and triple staining of eGFP-labeled fibers (green), synaptophysin-immunoreactive terminals (red) and FG-labeled mPFC-projecting neurons (blue) in the CM. The white rectangular is enlarged, and the white arrowheads indicate the eGFP-labeled fibers expressing synaptophysin which are in close contact with FG-labeled neurons in CM. Each experiment was repeated for 3 times independently with similar results. Scale bars = 100 µm (**f**, upper row in **g**), 10 µm (lower row in **g**). CM: central medial thalamic nucleus, D3V: dorsal 3rd ventricle, PVT: paraventricular thalamic nucleus.

into the ipsilateral mPFC (Fig. 5a, b). When applied with 473 nm laser light at 5, 10, and 20 Hz, the evoked excitatory postsynaptic currents (eEPSC) was consistent with the frequency of stimulation, implying effective ChR2 expression, and repeatedly proving the existence of the PBN-CM-mPFC pathway (Fig. 5c). Meanwhile, eEPSC were blocked with bath application of CNQX, an antagonist of the AMPA receptor, which suggests that the CM receives glutamatergic projections from the PBN (Fig. 5d, e).

Furthermore, the electrophysiological characteristics of mPFC-projecting CM neurons in response to acute and chronic itch stimuli were examined. Initially, both the amplitude and frequency of spontaneous EPSC (sEPSC) increased significantly in the DNFB group, but not in the His-injected group (Fig. 5f–h). Meanwhile, the parameters of passive membrane properties, membrane input resistance (Rin) and resting membrane potential (RMP), were significantly different between the DNFB and Acetone groups (Fig. 5i, j). In the current-clamp mode, we found that the minimum current required to elicit an action potential (Rheobase) decreased dramatically in the DNFB group, yet the action potential (AP) amplitude remained unchanged (Fig. 5k–m). The spike number induced by the depolarized step current was increased in mice performed with acute and chronic itch models. It is worth noticing that the spike number in the DNFB group was significantly higher than that in the His group, indicating a differently elevated excitability of the mPFC-projecting CM neurons with His and DNFB stimuli (Fig. 5n, o).

### Dynamics of the CM-mPFC pathway in different itch models
To further verify the subpopulation of neurons receiving CM projections, $AAV_{2/9}$-hSyn-eGFP-Synaptophysin-mRuby was injected into the CM (supplementary Figs. 4e–g). eGFP/mRuby double-labeled terminals were in close contact with both pyramidal neurons (PNs; using CaMKII as a marker) and GABAergic interneurons (INs; using GAD67 as

a pan-inhibitory neuronal marker) (Supplementary Figs. 4h, i). These results suggest that the CM sends projections to both PNs and INs in the mPFC.

To assess if the functional link between the CM and mPFC was constant or dynamic in acute and chronic itch models, electrophysiological methods were used. PNs were recognized by their morphological characteristics and intrinsic properties according to a previous report[19]. A virus using mdlx as a promoter ($AAV_{2/9}$-mdlx-eGFP), reported to specifically label cortical GABAergic neurons[20,21], was injected into the bilateral mPFC to label GABAergic neurons (Fig. 6a). The efficacy of this virus is confirmed in Supplementary Figs. 5a, b. In addition, the membrane capacitance (Cm) and other electrophysiological characteristics were also used to confirm the recorded $IN^{mPFC}$.

We then patched these two types of neurons and tested $CM-PN^{mPFC}$ and $CM-IN^{mPFC}$ connections in different itch models. $AAV_{2/9}$-ChR2 was injected into the CM, and the ChR2-containing CM-mPFC-projecting fibers were activated by optic stimulation through a ×40 objective lens over the field of the patched cells (Fig. 6a, b). Although no significant changes were observed in the Saline and Acetone groups, the amplitude of optic stimulation evoked EPSC in PNs was significantly higher in the His group, and lower in the DNFB group (Fig. 6c). Furthermore, the recorded PNs were injected with a two-phase 100 pA current (2 s duration) at a 2 s interval. Blue laser stimulation (1 ms, 20 Hz, 2-3 mW/mm²) was administered during the second train of depolarized currents to examine the overall effect of CM projections in different itch models (Fig. 6d). The photoactivation generated an increasing number of spikes in the His group and a significant decrease of spiking in the DNFB group (Fig. 6e). When summarizing the changes in spiking in five sweeps, the number of spikes was significantly lower in the DNFB group, and the His group displayed an increasing trend, though not significant (Fig. 6f). Based on these data, we hypothesized that CM projections might functionally give

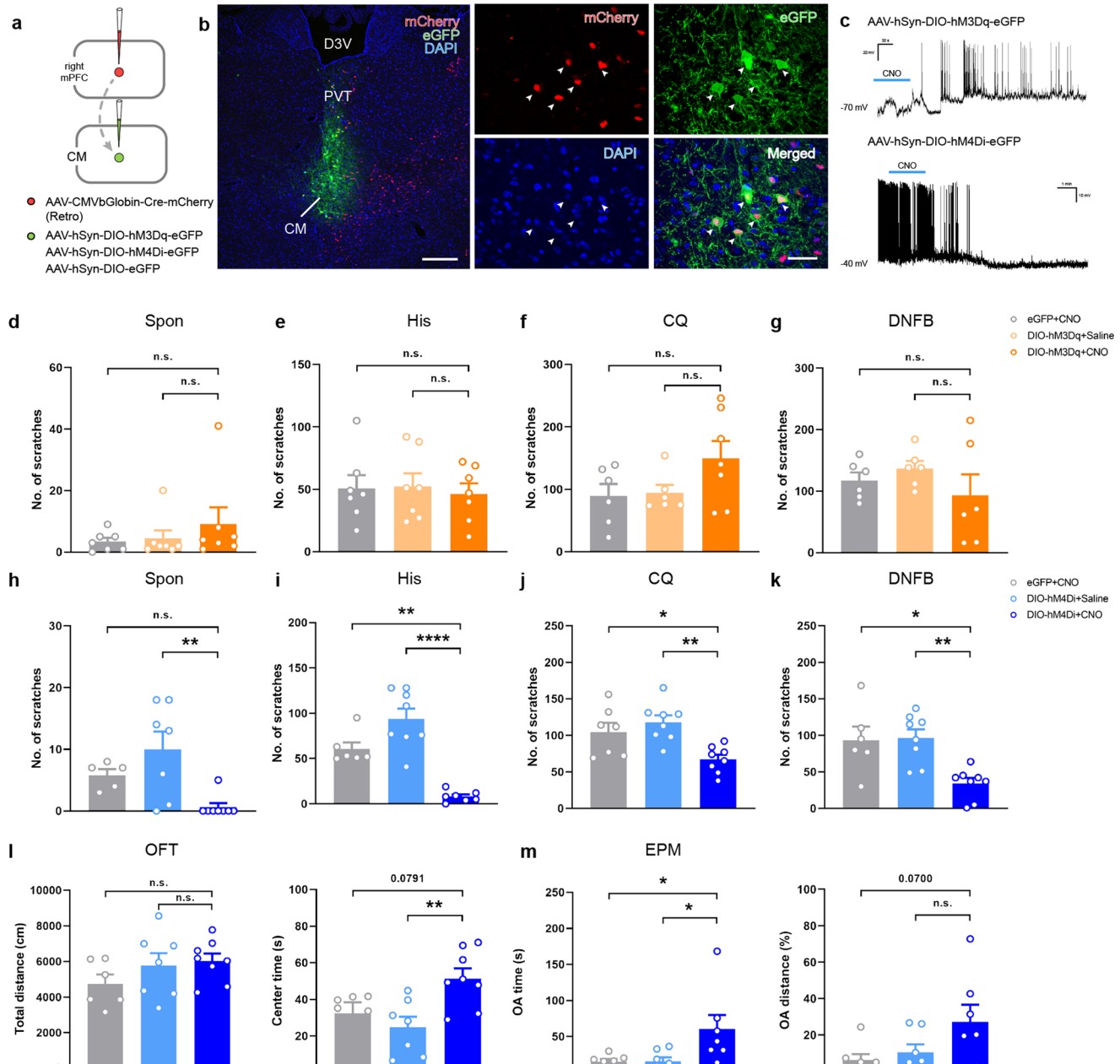

**Fig. 4 | Silencing the CM-mPFC pathway attenuates both scratching and chronic itch-induced anxiety-like behavior. a** Schematic indicating the virus strategy for pathway-specific excitation or inhibition. **b** Representative images depicting the localization of mCherry (red)/eGFP (green)/DAPI (blue) triple-labeled neurons (white arrowheads) in CM. Scale bars = 200 μm (left), 40 μm (right). **c** Bath application of CNO (10 μM) elicited neuronal discharge in neurons expressing hM3Dq-eGFP (clamped at −70 mV), whereas suppressed the spiking in hM4Di-injected group (clamped at −40 mV). **d, g** The modulatory effect of CNO in spontaneous, acute and chronic itch groups in chemogenetic activation of the CM-mPFC pathway (Spon: F(2,18) = 0.7521, P = 0.4856, CNO (eGFP vs. hM3Dq): P = 0.498, hM3Dq (Saline vs. CNO): P = 0.6184; n = 7 per group; His: F(2,18) = 0.09881, P = 0.9064, CNO (eGFP vs. hM3Dq): P = 0.9465, hM3Dq (Saline vs. CNO): P = 0.9042; n = 7 per group; CQ: F(2,16) = 2.440, P = 0.1189, CNO (eGFP vs. hM3Dq): P = 0.1532, hM3Dq (Saline vs. CNO): P = 0.2005; n = 6, 6, 7 per group; DNFB: F(2,15) = 0.9779, P = 0.3988, CNO (eGFP vs. hM3Dq): P = 0.7206, hM3Dq (Saline vs. CNO): P = 0.3681; n = 6 per group). **h, k** The ameliorated scratching behavior in spontaneous, acute and chronic itch models in inhibitory chemogenetic modulation group. (Spon:

F(2,17) = 7.191, P = 0.0055, CNO (eGFP vs. hM4Di): P = 0.1709, hM4Di (Saline vs. CNO): P = 0.0041; n = 5, 7, 8 per group; His: F(2,18) = 28.48, P < 0.0001, CNO (eGFP vs. hM4Di): P = 0.0012, hM4Di (Saline vs. CNO): P < 0.0001; n = 6, 8, 7 per group; CQ: F(2,20) = 7.949, P = 0.0029, CNO (eGFP vs. hM4Di): P = 0.0318, hM4Di (Saline vs. CNO): P = 0.0027; n = 7, 8, 8 per group; DNFB: F(2,19) = 8.485, P = 0.0023, CNO (eGFP vs. hM4Di): P = 0.0105, hM4Di (Saline vs. CNO): P = 0.0039; n = 6, 8, 8 per group). **l** Summary data of total distance and center time in mice performed with DNFB model in OFT (Total distance: F(2,18) = 1.400, P = 0.2723, CNO (eGFP vs. hM4Di), P = 0.2636, hM4Di (Saline vs. CNO): P = 0.9367; Center time: F(2,18) = 6.081, P = 0.0096, CNO (eGFP vs. hM4Di): P = 0.0791, hM4Di (Saline vs. CNO): P = 0.0091; n = 6, 7, 8 per group). **m** Summary data of mice with DNFB model in EPM (OA time: F(2,18) = 4.753, P = 0.0220, CNO (eGFP vs. hM4Di): P = 0.038, hM4Di (Saline vs. CNO): P = 0.041; OA distance: F(2,18) = 3.165, P = 0.0664, CNO (eGFP vs. hM4Di): P = 0.07, hM4Di (Saline vs. CNO): P = 0.1681; n = 7 per group). n.s.: no significance, *P < 0.05, **P < 0.01, ****P < 0.0001. Data are presented as mean ± S.E.M. Two-way ANOVA with Tukey's multiple comparison tests were used for **d–m**. D3V: dorsal 3rd ventricle.

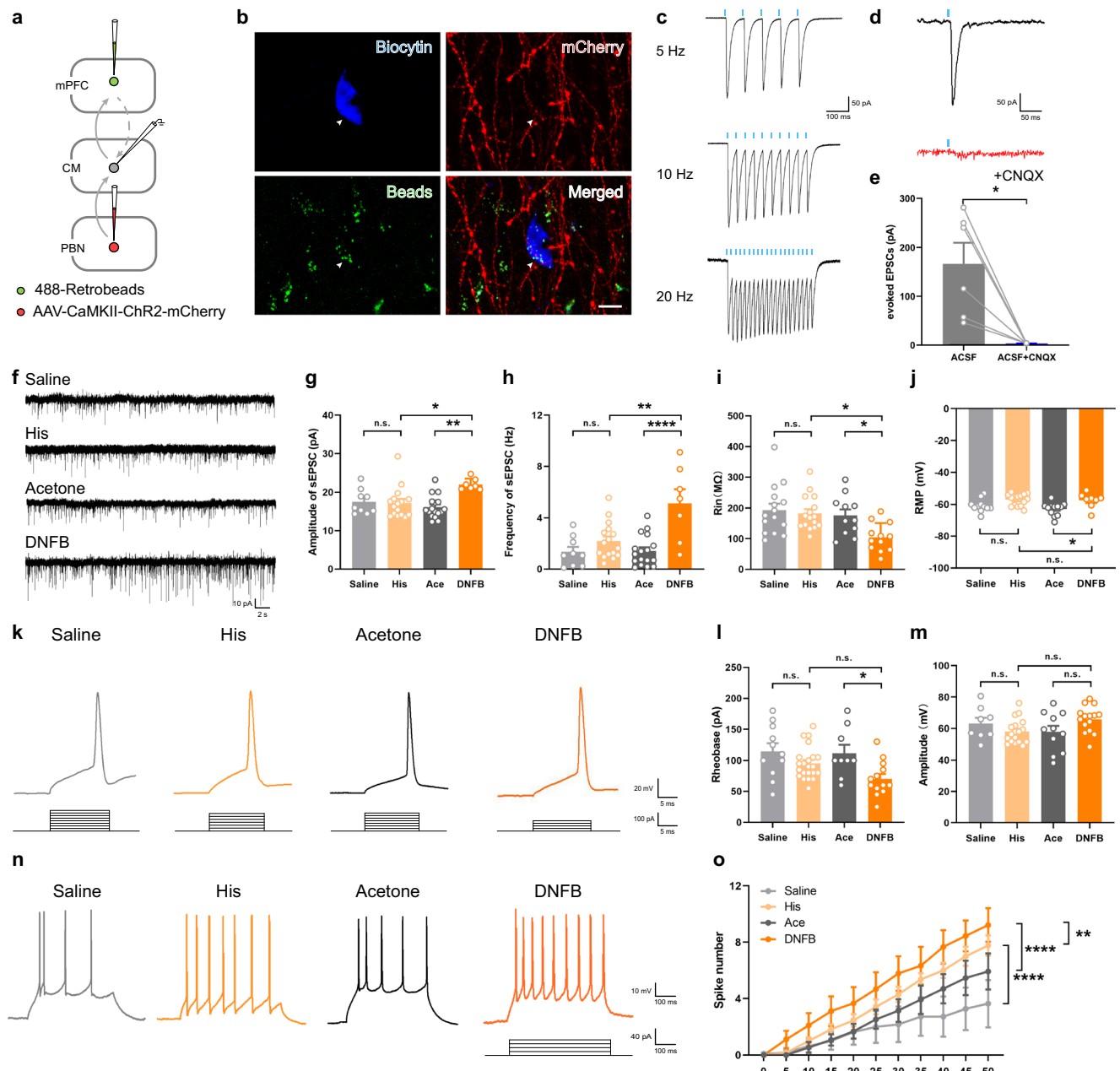

**Fig. 5 | Neuronal excitability of CM in acute and chronic itch models.**
**a** Schematic diagram of virus strategy, and the recording configuration in CM.
**b** Recorded neurons (biocytin-labeled, blue) were Retrobeads-positive (green), and surrounded by mCherry-labeled fibers and terminals (red) from the PBN (repeated in each successfully-recorded neuron). White arrowheads indicate synaptic connection with the recorded neuron. Scale bar = 10 μm. **c** Sample traces of the evoked excitatory postsynaptic currents (eEPSC) induced by blue light (473 nm, 2 ms, blue bars) in 5, 10, 20 Hz. **d, e** Representative traces and summary data depicting eEPSC evoked by photostimulation without or with CNQX (25 μM) application ($P = 0.0127$, $n = 6$ neurons from 3 mice). **f** Sample traces of spontaneous EPSC (sEPSC).
**g, h** Summary data of the amplitude and frequency of sEPSC (Amplitude: $F_{(3,45)} = 5.612$, $P = 0.0023$, Saline vs. His: $P = 0.9968$, Ace vs. DNFB: $P = 0.0011$, His vs. DNFB: $P = 0.0106$; $n = 9, 16, 17, 7$ neurons per group; Frequency: $F_{(3,45)} = 10.16$, $P < 0.0001$, Saline vs. His: $P = 0.5804$, Ace vs. DNFB: $P < 0.0001$, His vs. DNFB: $P = 0.001$; $n = 9, 16, 17, 7$ neurons per group). **i, j** Summary data of input resistance (Rin) and resting membrane potential (RMP) (Rin: $F_{(3,50)} = 5.050$, $P = 0.0039$, Saline vs. His: $P = 0.9663$, Ace vs. DNFB: $P = 0.0478$, His vs. DNFB: $P = 0.0125$; $n = 15, 16, 11, 12$ neurons per group; RMP: $F_{(3,42)} = 5.673$, $P = 0.0024$, Saline vs. His:

$P = 0.1165$, Ace vs. DNFB: $P = 0.0244$, His vs. DNFB: $P = 0.9976$; $n = 11, 15, 12, 8$ neurons per group). **k** Sample traces of action potential induced by step-wise depolarized current (15 ms; 5 pA increments). **l, m** Summary data of the rheobase and the AP amplitude (Rheobase: $F_{(3,51)} = 4.591$, $P = 0.0064$, Saline vs. His: $P = 0.382$, Ace vs. DNFB: $P = 0.0252$, His vs. DNFB: $P = 0.1409$; $n = 11, 22, 9, 13$ neurons per group; AP amplitude: $F_{(3,48)} = 2.414$, $P = 0.0781$, Saline vs. His: $P = 0.5873$, Ace vs. DNFB: $P = 0.1641$, His vs. DNFB: $P = 0.1014$; $n = 8, 18, 11, 15$ neurons per group).
**n** Representative traces of firing pattern while recording mPFC-projecting CM neurons (500 ms; 40 pA). **o** Summary data of the spike number induced by stepwise depolarized currents (500 ms; 5 pA increments; interaction: $P = 0.5909$, treatment: $P < 0.0001$, Saline vs. His: $P < 0.0001$, Ace vs. DNFB: $P < 0.0001$, His vs. DNFB: $P = 0.0056$; Saline: $n = 11, 17, 13, 9$ neurons. The number of mice were 3, 4, 4, 4 in Saline, His, Ace and DNFB group in **g–j, l, m** and **o**, respectively). n.s.: not significant, \*$P < 0.05$, \*\*$P < 0.01$, \*\*\*$P < 0.0001$. Data are presented as mean ± S.E.M. Two-tailed, paired, Student's *t*-test for **e**, one-way ANOVA with Tukey's multiple comparison tests for **g–j** and **l, m** and two-way ANOVA with Tukey's multiple comparison tests for **o**.

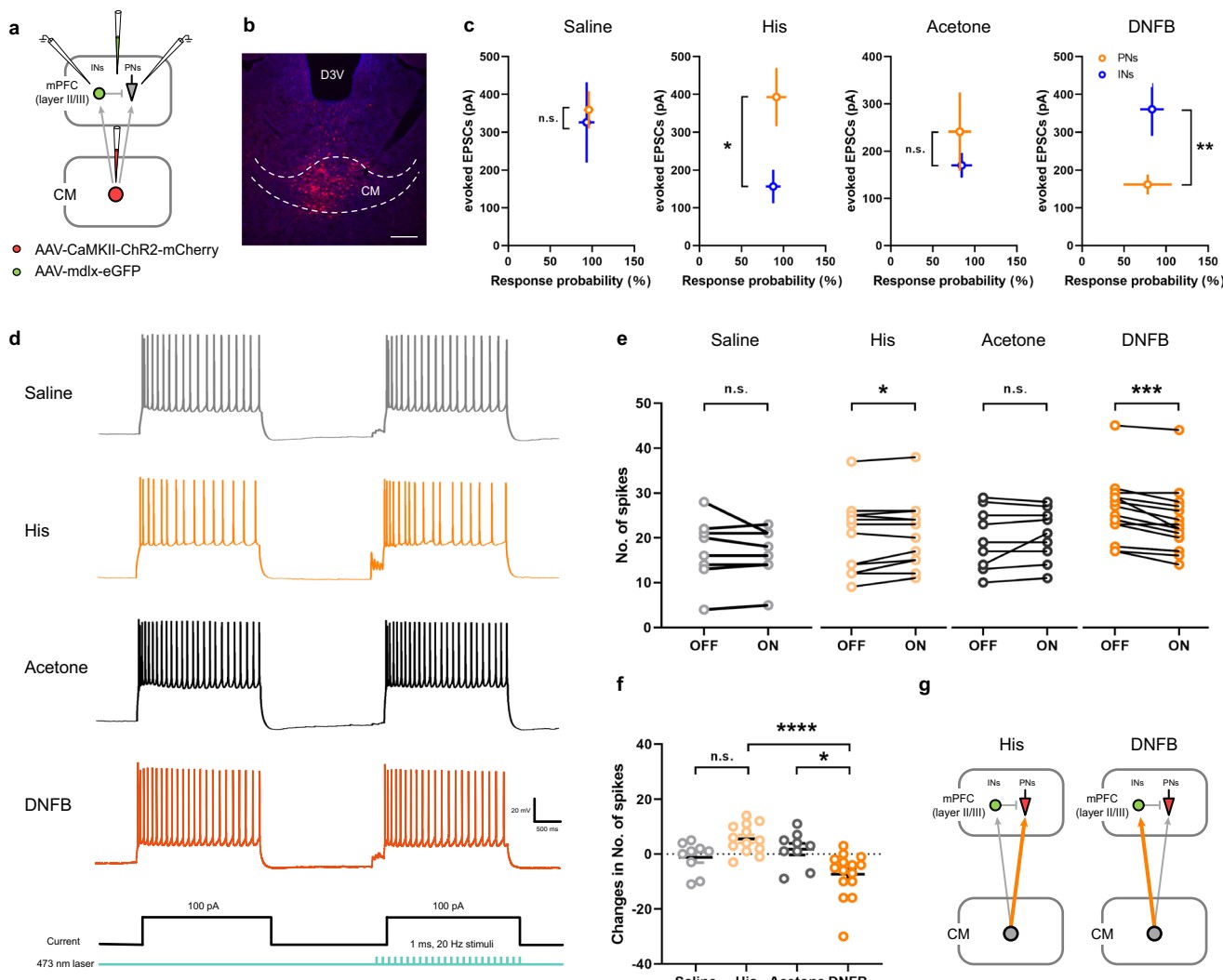

**Fig. 6 | Dynamic CM projections in acute and chronic itch models. a** Schematic diagram of virus injection into the CM, and the recording configuration of pyramidal neurons (PNs) or GABAergic interneurons (INs) in the mPFC. **b** Representative image of ChR2-mChery injection into the CM (white dashed lines). Scale bar = 200 μm. **c** The connectivity strength of CM projections to PNs and INs in Saline, His, Acetone and DNFB groups with blue light (473 nm, 2 ms) stimulation (Saline: $P = 0.7677$, $n = 8$ INs, 10 PNs from 4 mice; His: $P = 0.0172$, $n = 11$ INs, 12 PNs from 6 mice; Ace: $P = 0.4253$, $n = 9$ INs, 9 PNs from 5 mice; DNFB: $P = 0.0067$, $n = 12$ INs, 16 PNs from 7 mice). **d** Example traces depicting spiking number of PNs in response to two depolarizing currents injections with or without optogenetic stimulation (473 nm, 1 ms, 20 Hz) as depicted in the protocol (bottom). **e** Number of spikes in PNs plotted during light off and light on periods. There was significant increase in spiking number in His group during light on period, and the number of

spikes decreased dramatically in DNFB group after photoactivation (Saline: $P = 0.4554$, $n = 9$ from 3 mice; His: $P = 0.0395$, $n = 13$ from 4 mice; Ace: $P = 0.3027$, $n = 9$ from 3 mice; DNFB: $P = 0.0002$, $n = 15$ from 4 mice). **f** Changes in the number of spikes summarized from 5 sweeps during light off and light on periods (F(3,42) = 9.261, $P < 0.0001$, Saline vs. His: $P = 0.1059$, Ace vs. DNFB: $P = 0.0114$, His vs. DNFB: $P < 0.0001$; Saline: $n = 9$ from 3 mice; His: $n = 13$ from 4 mice; Ace: $n = 9$ from 3 mice; DNFB: $n = 15$ from 4 mice). **g** A hypothesized connection patten of preferential CM projections to PN^mPFC and IN^mPFC (Orange lines) in acute and chronic itch models, respectively. n.s.: no significance, $^*P < 0.05$, $^{**}P < 0.01$, $^{***}P < 0.001$, $^{****}P < 0.0001$. Data are presented as mean ± S.E.M. Two-tailed, unpaired, Student's $t$-test for **c**, Two-tailed, paired, Student's $t$-test for **e** and one-way ANOVA with Tukey's multiple comparison tests for **f**.

predominance to PNs in the His model, which transfers to INs in the DNFB model (Fig. 6g).

The possible mechanism was further verified using a similar viral strategy, according to the timeline (Fig. 7a, b). It was noted that eEPSC could be induced by blue laser light when either PNs or INs were recorded. eEPSC were blocked by tetrodotoxin (TTX; a sodium channel blocker) and then rescued by application of 4-aminopyridine (4-AP; a potassium channel blocker), which could cause calcium influx that facilitates vesicular release when photoactivating ChR2-expressing terminals[22]. These results indicate that both PNs and INs receive monosynaptic inputs from the CM. However, the evoked inhibitory postsynaptic currents (eIPSC) in PNs disappeared after TTX application and could not be rescued by 4-AP. These results indicate

disynaptic feedforward inhibition in PNs that is mediated by INs (Fig. 7c, d). Feedforward inhibition was also demonstrated by a significantly longer onset latency of eIPSC than that of eEPSC (Fig. 7e, f). While no significant changes were observed between the Saline and His groups in the eIPSC/eEPSC ratio, the ratio increased dramatically in the DNFB group, suggesting an E/I imbalance induced by CM projections under chronic itch condition (Fig. 7g). Concomitantly, variations were observed in the paired pulse ratio (PPR), an index that is inversely related to neurotransmitter release probability, in both the His and DNFB groups. The results suggested a His-induced higher connectivity from the CM to the PN^mPFC. Additionally, the PPR of both eEPSC and eIPSC decreased significantly in DNFB model, which indicated the higher release probability of CM-PN^mPFC projections, as well as a

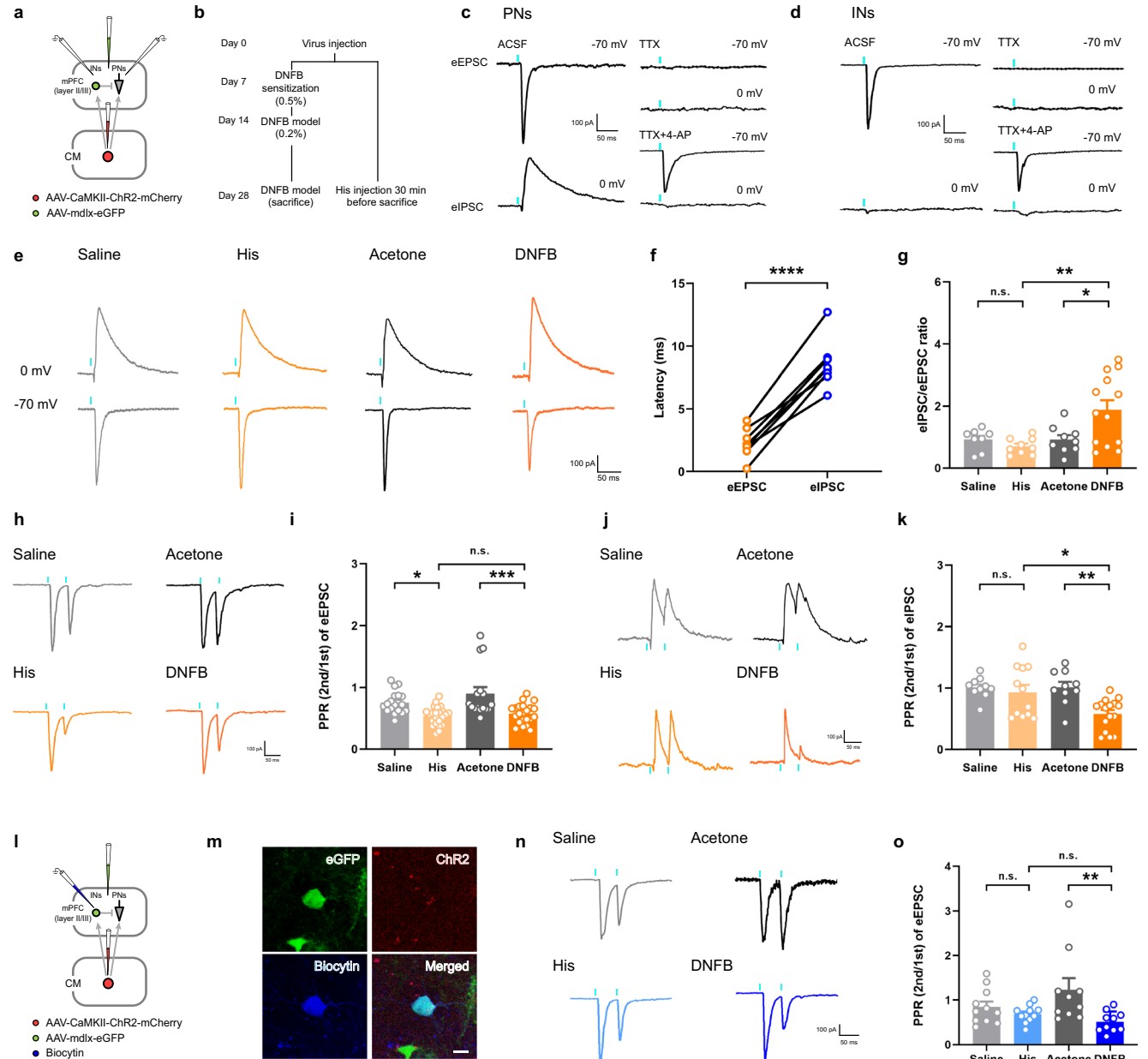

**Fig. 7 | An enhanced feedforward inhibition in chronic itch model. a** Schematic diagram of virus strategy and the recording configuration of PN$^{mPFC}$ or IN$^{mPFC}$. **b** Mice were performed with acute and chronic itch models for electrophysiological recording according to the time line. **c, d** Sample traces of blue light (473 nm, 2 ms, blue bars)-induced eEPSC and evoked inhibitory postsynaptic currents (eIPSC) in PNs, without or with tetrodotoxin (TTX, 1 μM) or the combination of TTX and 4-aminopyridine (4-AP, 100 μM). **e** Representative traces of eIPSC (voltage clamped at 0 mV) and eEPSC (voltage clamped at −70 mV) in PNs in response to photostimulation. **f** Summary data of eEPSC and eIPSC latency to onset of blue-laser stimulation (eEPSC: $n = 8$ neurons from 4 mice; eIPSC: $n = 8$ neurons from 7 mice, $P < 0.0001$). **g** Statistical analysis of eIPSC/eEPSC ratio in Saline, His, Acetone and DNFB groups ($F_{(3,35)} = 6.182$, $P = 0.0017$, Saline vs. His: $P = 0.9229$, Ace vs. DNFB: $P = 0.0195$, His vs. DNFB: $P = 0.0032$; Saline: $n = 8$ from 4 mice; His: $n = 9$ from 5 mice; Ace: $n = 9$ from 4 mice; DNFB: $n = 13$ from 5 mice). **h, i** Representative traces and summary data of PNs-mediated paired-pulse eEPSC induced by the paired light pulses (473 nm, 2 ms) given at 50 ms interval ($F_{(3,75)} = 9.531$, $P < 0.0001$,

Saline vs. His: $P = 0.0245$, Ace vs. DNFB: $P = 0.0004$, His vs. DNFB: $P = 0.9583$; Saline: $n = 18$ from 4 mice; His: $n = 23$ from 5 mice; Ace: $n = 16$ from 4 mice; DNFB: $n = 22$ from 5 mice). **j, k** Representative traces and summary data of eIPSC with paired light stimulation given at 50 ms interval (voltage clamped at 0 mV). Quantitative analysis exhibited a dramatic decrease of PPR in DNFB group ($F_{(3,44)} = 6.720$, $P = 0.0008$, Saline vs. His: $P = 0.93$, Ace vs. DNFB: $P = 0.004$, His vs. DNFB: $P = 0.0164$; Saline: $n = 10$ from 4 mice; His: $n = 12$ from 5 mice; Ace: $n = 10$ from 4 mice; DNFB: $n = 16$ from 5 mice). **l** Schematic diagram of virus strategy and the recording configuration of IN$^{mPFC}$. **m** Representative images of recorded interneuron (Biocytin-labeled, blue; repeated in each successfully-recorded neuron). Scale bar = 10 μm.
**n, o** Representative traces and summary data of INs-mediated PPR ($F_{(3,38)} = 4.187$, $P = 0.0118$, Saline vs. His: $P = 0.8564$, Ace vs. DNFB: $P = 0.0088$, His vs. DNFB: $P = 0.836$; Saline: $n = 11$ from 4 mice; His: $n = 11$ from 5 mice; Ace: $n = 10$ from 5 mice; DNFB: $n = 10$ from 6 mice). n.s.: no significance, $^*P < 0.05$, $^{**}P < 0.01$, $^{***}P < 0.001$, $^{****}P < 0.0001$. Data are presented as mean ± S.E.M. Two-tailed, paired, Student's $t$-test for **f**, and one-way ANOVA with Tukey's multiple comparison tests for **g, i, k, o**.

strengthened feedforward inhibition input from the IN$^{mPFC}$ with chronic itch stimuli (Fig. 7h–k). While recording the labeled INs in the mPFC, the significantly lower PPR of eEPSC in the DNFB group also indicated enhanced CM-IN$^{mPFC}$ connections in chronic itch model

(Fig. 7l–o). These data support the hypothesis that acute itch stimuli induce an elevated release probability from the CM presynaptic terminal to the PN$^{mPFC}$, whereas the chronic itch model enhanced the CM connections to the IN$^{mPFC}$.

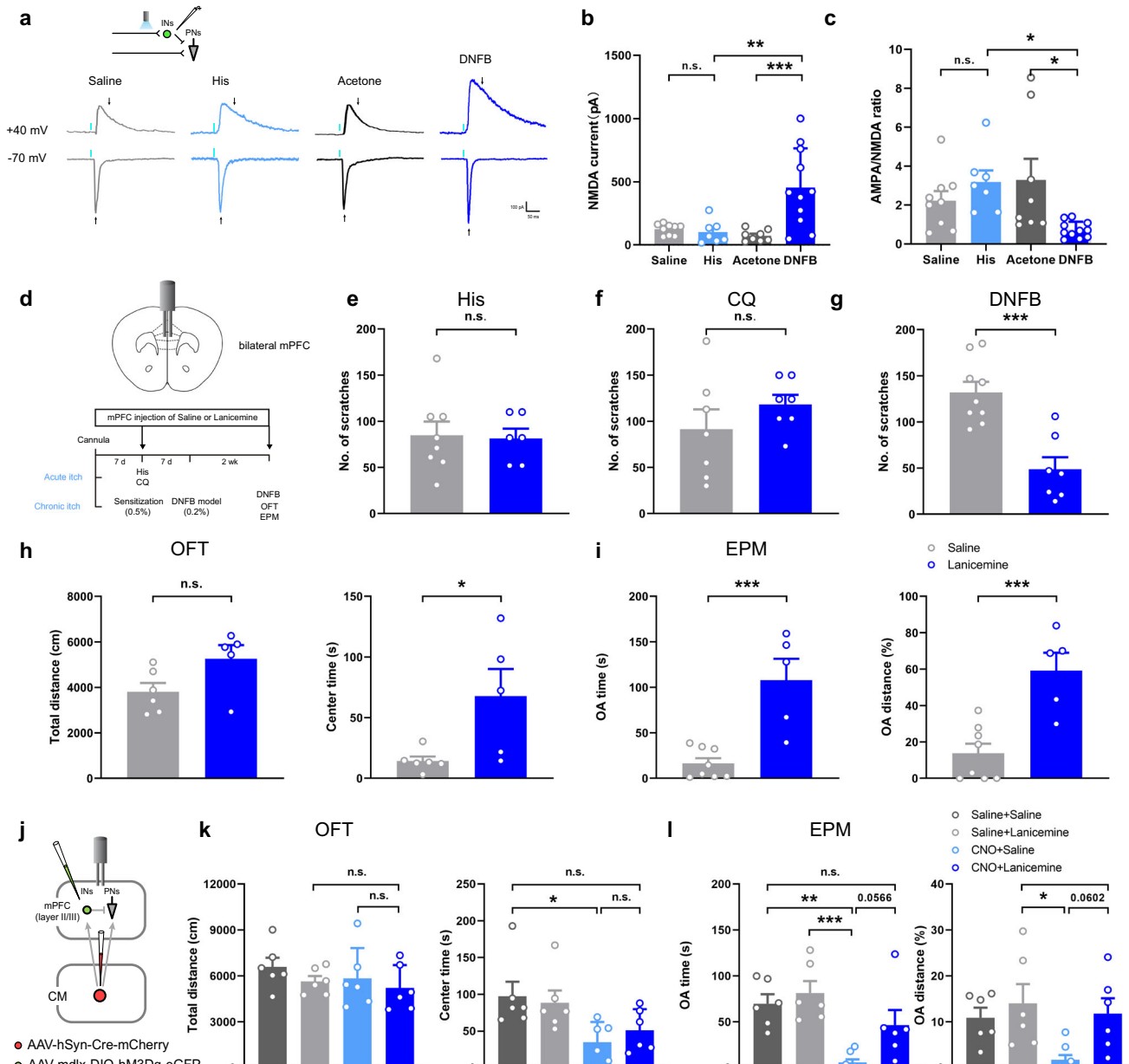

**Fig. 8 | NMDA receptor as a potential target for behavioral and affective modulation in chronic itch. a** Schematic diagram and representative traces of blue light evoked NMDA (voltage clamped at 40 mV) and AMPA currents (voltage clamped at −70 mV) in IN^mPFC. Black arrows indicate the time points of current amplitude used for further analysis. **b, c** Summary data of NMDA current and AMPA/NMDA ratio in acute and chronic itch groups (NMDA current: F(3,31) = 9.439, $P = 0.0001$, Saline vs. His: $P = 0.9953$, Ace vs. DNFB: $P = 0.0005$, His vs. DNFB: $P = 0.0022$; AMPA/NMDA: F(3,31) = 4.088, $P = 0.0148$, Saline vs. His: $P = 0.7221$, Ace vs. DNFB: $P = 0.023$, His vs. DNFB: $P = 0.0404$; Saline: n = 9 from 4 mice; His: n = 7 from 5 mice; Ace: $n = 8$ from 5 mice; DNFB: $n = 11$ from 6 mice). **d** Pharmacological manipulations with cannula implanted into bilateral mPFC (top). Saline and Lanicemine applications were performed according to the timeline (bottom). **e–g** Summary data of scratching behavior in acute itch and chronic itch models (His: $P = 0.8587$; $n = 8$, 6 per group; CQ: $P = 0.2846$; $n = 7$ per group; DNFB: $P = 0.0003$; $n = 9$, 7 per group). **h** Lanicemine application significantly increased the center time of mice performed with DNFB model without affecting the total distance in OFT (Total distance: $P = 0.0632$; Center time: $P = 0.029$; $n = 6$, 5 per group). **i** Summary

data of EPM after Lanicemine administration into bilateral mPFC (OA time: $P = 0.0006$; OA distance: $P = 0.0009$; $n = 8$, 5 per group). **j** Schematic diagram of virus strategy in specific activation of the CM-IN^mPFC pathway, and the cannula implantation in the mPFC. **k, l** Summary data of OFT and EPM after CNO and/or Lanicemine application (Total distance: F(3,20) = 0.8828, $P = 0.4668$; mPFC^Lanicemine (Saline vs. CNO): $P = 0.9592$; CNO (mPFC^Saline vs. mPFC^Lanicemine): $P = 0.8901$; Center time: F(3,20) = 3.840: $P = 0.0254$; mPFC^Saline (Saline vs. CNO): $P = 0.0402$; CNO (mPFC^Saline vs. mPFC^Lanicemine): $P = 0.8753$; Saline-mPFC^Saline vs. CNO-mPFC^Lanicemine: $P = 0.1705$; $n = 6$ per group; OA time: F(3,22) = 10.37, $P = 0.0002$, mPFC^Saline (Saline vs. CNO): $P = 0.0016$; Saline-mPFC^Lanicemine vs. CNO-mPFC^Saline: $P = 0.0002$; CNO (mPFC^Saline vs. mPFC^Lanicemine): $P = 0.0566$; Saline-mPFC^Saline vs. CNO-mPFC^Lanicemine: $P = 0.4733$; OA distance: F(3,22) = 4.467, $P = 0.0135$, Saline-mPFC^Lanicemine vs. CNO-mPFC^Saline: $P = 0.0164$; CNO (mPFC^Saline vs. mPFC^Lanicemine): $P = 0.0602$; mPFC^Lanicemine (Saline vs. CNO): $P = 0.9419$; $n = 6$, 6, 8, 6 per group). n.s.: no significance, *$P < 0.05$, **$P < 0.01$, ****$P < 0.0001$. Data are presented as mean ± S.E.M. One-way ANOVA with Tukey's multiple comparison tests for **b, c** and **k, l**. Two-tailed, unpaired, Student's $t$-test for **e–i**.

Further, we tested the AMPA and NMDA currents in the PN^mPFC and IN^mPFC. In the voltage-clamp mode, the NMDA amplitude of the IN^mPFC was significantly increased in the DNFB group, and the AMPA/NMDA ratio decreased dramatically (Fig. 8a–c). To further ensure the

variations of IN^mPFC in different itch models, GAD2-eGFP mice, with specific labeling of GABAergic neurons, were also performed with similar whole-cell recordings, which showed the same trend in PPR, AMPA, NMDA and NMDA/AMPA ratio as that in mdlx-eGFP injected

mice (supplementary Figs. 5c–g). However, the DNFB stimuli exhibited no influence on either the amplitude of the NMDA current or the AMPA/NMDA ratio of the PN[mPFC] (supplementary Fig. 6), indicating that elevated NMDA current in the IN[mPFC] might be specifically induced by the DNFB model.

## Lanicemine in relieving itch sensation and chronic itch-induced anxiety-like behavior

NMDA receptor antagonists have been reported to exhibit analgesic and antidepressant effects in the mPFC[20,23]. Since that the NMDA current is enhanced only in the chronic itch model, we examined whether inhibition of NMDA receptors would reverse the scratching behavior and chronic itch-induced affective behavior. A low-trapping NMDA antagonist, Lanicemine, was then chosen for pharmacological manipulations to avoid psychotomimetic side effects relative to ketamine, another frequently used antidepressant[24]. A bilateral cannula was implanted into the mPFC, and acute and chronic itch models were established according to the timeline (Fig. 8d). Lanicemine exhibited no modulatory effect on the number of scratches in the His and CQ groups; however, it significantly attenuated the scratching behavior in the DNFB group (Fig. 8e–g). In chronic itch induced anxiety-like behavior, it was found that center time was elevated, and that OA time and OA distance also exhibited a significant increase in the Lanicemine-treated group (Fig. 8h, i). Combined with the results above, Lanicemine attenuated both scratching behaviors and affective behavior induced in the chronic itch model, probably by antagonizing NMDA receptors. To further examine the role of NMDA receptor, the virus AAV$_{2/9}$-CMV-DIO-(EGFP-U6)-shRNA (NR1) (with AAV$_{2/9}$-CMV-DIO-(EGFP-U6)-shRNA (scramble) as control) was also designed to specifically knockdown the NR1 subunit of NMDA receptors. Combined with GAD2-Cre mice, such virus strategy could specifically knockdown the NMDA receptor on the IN[mPFC]. Results indicated that IN[mPFC] NMDA receptor NR1 knockdown did not effect the Spontaneous, His- or CQ-induced scratching behavior. It, however, significantly reduced the scratching behavior in DNFB model, as well as relieved the chronic itch-induced anxiety-like behavior in both OFT and EPM tests (supplementary Fig. 7).

Furthermore, according to the electrophysiological results above, the CM-IN[mPFC] was activated to simulate the CM-mPFC pathway status in the chronic itch model (Fig. 8j). Although no significant changes were observed in the OFT, both OA time and OA distance exhibited an increasing trend in the CNO+Lanicemine group (Fig. 8k–l). These results imply that the enhanced connection from the CM to the IN[mPFC] might, at least partially, explain the anxiety-like behavior in the DNFB model, and that the NMDA receptor antagonist could reverse anxiety-like behavior.

## Discussion

In the present study, anatomical, chemogenetic, optogenetic, and electrophysiological methods were combined to corroborate the PBN-CM-mPFC pathway for itch signal transmission. Here, we defined a selective augmentation of CM input to PNs and INs in acute and chronic itch, respectively, and the distorted E/I balance induced by the elevated NMDA current of INs in the DNFB model, thereby explaining the mechanism of chronic itch-induced affective behavior. These results provide a potential target for clinical use in ameliorating scratching behavior and negative feelings simultaneously.

CM has been proven to connect with the brainstem and cortex, which lays the foundation for its abundant modulatory effects[9]. Functionally, CM is more closely related to affective pain processing, and has been used as a practical clinical target nucleus in deep brain stimulation for analgesia in intractable neuropathic pain[25,26]. A few previous imaging studies have mentioned the activation of the thalamus under itch stimuli. However, their findings cannot be narrowed down to CM owing to limited spatial resolution[27,28]. In the present study, such limitations were partially covered by fiber photometry and

electrolytic lesion results. In electrolytic lesions of CM, scratching behavior was dramatically relieved in both acute and chronic itch models, which is in accordance with a hypothetical itch network in which the thalamus acts as a central relay of itch signals[29]. Furthermore, in chronic itch-induced anxiety-like behavior, although no dramatic changes were observed in the lesion group, CM lesions exhibited a positive trend in both the OFT and EPM. Together with prior research that rostral CM is involved in the transmission of noxious feelings[25], the present study clarifies the participation of CM in pruriceptive sensation, be it a sensory or affective aspect.

PBN gathers the main sensory input, including itch signals, from the spinal cord[8]. It has also been demonstrated to transmit pain-related negative emotions to the intralaminar thalamic nucleus[30]. However, the functions of each specific subnucleus have not been specifically demonstrated[31]. The present work addressed this issue by injecting a tracing virus with an activity-dependent promoter. In lines with a previous retrograde tracing study on dMITC[32], these morphological results confirmed the existence of a glutamatergic PBN-CM itch transmission pathway. On the other hand, photoinhibition of the PBN-CM pathway indicated that CM is the effluent nucleus of PBN, which is involved in itch signal transmission.

The prefrontal cortex has been implicated in both acute and chronic itch models[6]. This has been earlier proven by increased blood oxygen level-dependent levels or higher functional connectivity in clinical imaging studies[33–35]. It is also activated in nocebo-induced itch and contagious itch, and is regarded as the central hub in mediating the cognitive and motivational aspects of itch[36,37]. The present study has proved the existence of a PBN-CM-mPFC pathway, similar to previous morphological evidence[9,38]. Intriguingly, our results of CM-mPFC chemogenetic activation did not reveal any behavioral changes, whereas inhibiting the CM-mPFC pathway was sufficient to relieve scratching behavior. This indicates that the CM-mPFC pathway is at least one of the necessary pathways for itch signal transmission at the supraspinal level of the central nervous system. it has also been hypothesized that the mPFC participates in the control of scratching responses (pleasure/reward aspects) in pruriceptive sensation[29,39], and that the CM is required in attention-related tasks[40,41]. Therefore, the CM-mPFC pathway is likely to be involved in itch attention, which is consistent with the fact that itching acts as an alarm signal against environmental insults. In nociceptive sensation, lesions in the mPFC could significantly attenuate the affective behavior induced by the neuropathic pain model[20,42]. Silencing the CM-mPFC pathway could reverse anxiety-like behavior induced by the chronic itch model, which was not observed in the CM lesion experiment. These results may be explained by the selective role of mPFC-projecting CM neurons in the chronic itch model, and the fact that the relieved anxiety-like behavior is more likely to be a pathway-specific change.

Acute itching transmits alerting information of insults from the surroundings, whereas chronic itching generates pathological stimulation that triggers negative emotional outcomes[43], which might be related with distinct subpopulation of neurons[44–46]. Since dMITC subnuclei, such as the mediodorsal thalamic nucleus (MD) and paraventricular thalamic nucleus (PVT), have been shown to respond to different external stimuli through dynamic connections, either by changing neuronal excitabilities or by alternating the target nuclei[47,48], we hypothesized that the CM-mPFC pathway might be differently recruited in acute and chronic itch models. It has been validated that layers II/III of the mPFC receive long-range inputs from a number of structures for further integration of information[14,49]. Given that abundant fibers and terminals could be observed in layers II/III from the CM, these layers were chosen as the targets in electrophysiological experiments.

The hypothesis was verified by cell-type-specific recording of PN[mPFC] and IN[mPFC] in different models. Our results suggest a strengthened CM input to PN[mPFC] in the acute itch model, whereas projections

to IN[mPFC] were enhanced in the chronic itch model. According to unpublished data from our laboratory, activation of CaMKII-positive neurons could result in an increase in scratching behavior. Therefore, we speculated that the priority of CM input to PN[mPFC] in the acute itch model is to carry sensory information about external insults. The main afferent nuclei of the PFC, MD and PVT have been proven to modulate negative emotions by connecting with parvalbumin-positive (PV+) and nNOS-expressing GABAergic neurons, respectively[50,51]. Since the activation of GABAergic neurons could directly induce affective behaviors[20], and that inhibiting GABAergic neurons in the mPFC is sufficient and necessary to exhibit a significant antidepressant effect[23,50], the preference of CM input to IN[mPFC] in the DNFB group might be related to chronic itch-induced anxiety-like behavior. It has been documented that the brain processes different excitabilities and functional connectivity in acute and chronic itch models in response to environmental stimuli[13,33,52]. Our results on the CM dynamics in projections to the mPFC indicated a pathway-specific adaptation to different itch stimuli. It also implies that a rigid separation of the sensory and affective pathways of itch is over-simplistic, and the CM-mPFC pathway might participate in these two itch dimensions through different coding strategies.

The underlying mechanism of these dynamics could be, at least partially, explained by the changes in neuronal excitability of mPFC-projecting CM neurons and presynaptic inputs. In both acute and chronic itch models, the enhanced CM projections to PN[mPFC] is on account of both higher neuronal excitabilities of CM and elevated presynaptic transmission. The PPR of eEPSC in PN[mPFC] decreased in both His and DNFB groups, indicating the strengthened CM-PN[mPFC] connections in both acute and chronic itch models. However, the PPR of eIPSC in PN[mPFC] and the PPR of eEPSC in IN[mPFC] only decreased significantly in the DNFB model. Combined with the fact that the AMPA current of the PN[mPFC] decreased in DNFB model, we hypothesized that the activation from CM to the IN[mPFC] was too strong that overcame the effect of the CM-PN[mPFC] pathway activation. Additionally, the eEPSC/eIPSC ratio elicited by CM input shifted toward inhibition in the setting of the DNFB model, suggesting enhanced feedforward inhibition. Since it has been indicated that decreased excitatory/inhibitory (E/I) balance in the PFC is one of the most essential mechanisms of a series of mental disorders and chronic pain-induced negative emotions[20,53–55], the obtained results potentially explain the anxiety-like behavior induced by the DNFB model.

Another piece of evidence to verify the mechanism of dynamic modulations is that the NMDA current increased in IN[mPFC], yet remained unchanged when testing PN[mPFC] in the DNFB model. In the present study, we chose a low-trapping NMDAR antagonist, Lanicemine (a glutamate-targeting antidepressant) as a potential drug, which has almost no psychotomimetic properties or side effects when compared to ketamine[24,56]. The scratching behavior in the DNFB model and Chronic itch-induced anxiety-like behavior were both relieved after Lanicemine application, or by specifically knockdown of the NR1 subunit of NMDA receptor on IN[mPFC], which is in consistent with the NMDA current changes of the IN[mPFC] noted only in the chronic itch model. Although no clear evidence has been reported about the cell-type-specific target of Lanicemine, the modulatory effect of this drug on the behavioral and affective dimensions of chronic itch might indicate that this drug, like ketamine, might also primarily act on the NMDA receptors on IN[mPFC][23].

The present study highlighted CM as a platform for the therapeutic investigation of mPFC-dependent chronic itch-induced anxiety-like behavior. It has also delineated a brainstem-thalamus-cortex transmission pathway, which clarifies the central transmission pathway of itch signals after PBN receives spinal cord projections. Interestingly, this study elucidated a dynamic connection of CM projections to PN[mPFC] and IN[mPFC] with acute and chronic itch stimuli, respectively, suggesting a CM-mPFC pathway adjusting to a complex external

environment. However, the subclassification of IN[mPFC], such as SOM+ and PV+, has also been proven to differentially influence the modulatory effect of mPFC[50]. Therefore, additional studies are needed to specify the specific role of certain IN subpopulations in itch sensation and modulation, which could provide a clinical target for further use.

## Methods

All procedures have been approved by the Animal Care and Use Committees at The Fourth Military Medical University (Xi'an, China).

### Animals

A total number of 417 Adult male C57Bl/6 mice, 28 GAD2-Cre mice and 35 GAD2-eGFP mice (8 to 12 weeks) were used in the present study, weighing 20 to 35 g. All animals used were housed in a 12 h light/dark cycle with food and water given ad libitum, the ambient temperature (18 – 26 °C) and humidity (40 – 70%) have been controlled. All experiments were designed to use mice as few as possible according to the 3 R principle. All protocols involving mice have been optimized to minimize the suffering of animals from surgical operations.

### Brain stereotaxic injection

All animals were handled for 3 successive days ahead of brain stereotactic injection procedures to minimize the stress. Animals were deeply anesthetized with intraperitoneal (*i.p.*) injection of pentobarbital sodium (40 mg/kg), and then fixed on a stereotaxic apparatus (Narishige Scientific Instrument Lab, Tokyo, Japan). Mice used for morphological experiments were stereotactically injected with retrograde tracers or viral victors with a 1 μl Hamilton microsyringe connecting with a glass micropipette (internal tip diameter 15–25 μm) at a rate of 10 nl/min, and then kept in injection sites for 10 min before withdrawn to avoid drug diffusion. The injection coordinates are listed below based on the brain atlas[57]: PBN (AP: −5.20 mm; ML: − 1.30 mm; DV: − 3.60 mm); CM (AP: − 1.06 mm; ML: 0.00 mm; DV: − 3.50 mm); CM 20°(AP: − 1.06 mm; ML: − 1.70 mm; DV: − 3.92 mm); mPFC (AP: + 1.78 mm; ML: ± 0.25 mm; DV: − 2.50 mm).

All retrograde tracers and viruses concerning anatomical or behavioral assays were listed in supplementary table 1. For optogenetic manipulation, mice were implanted with optical fibers (NA 0.37, 200 μm, Shanghai Fiblaser Technology Co., Ltd, Shanghai, China) in CM (AP: − 1.06 mm; ML: 0.00 mm; DV: − 3.20 mm) 3 weeks after virus injection. After 1 week of recovery, mice were handled and habituated for 3 successive days, and then used for further behavioral experiments.

### Electrolytic lesion

Mice were anesthetized with pentobarbital sodium (40 mg/kg; i.p.), and then placed on the stereotaxic apparatus. The lesion of the CM was performed using a stainless-steel electrode with a DC current of 0.35 mA for 20 s with an electronic stimulator (SS-202J, Nihon Kohden, Tokyo, Japan) according to the above-mentioned coordinates. Mice with electrode inserted in CM yet did not pass any current were used as the sham-treated group. All behavioral assays were performed 7 d after the lesion.

### Fiber photometry

AAV[2/9] with CaMKII promotor and expressing GCaMP6m was injected into the CM to examine the neuronal dynamic of its glutamatergic neurons. Meanwhile, to test the participation of the PBN-CM pathway in itch sensation, anterograde virus that carries Cre enzyme to spread monosynaptically into the soma of neurons was injected into the PBN, and AAV[2/9] with double-floxed inverted open reading frame (DIO) expressing GCaMP7s was injected into the CM. Optic fiber was implanted into the CM (AP: −1.06 mm; ML: 0.00 mm; DV: − 3.50 mm) 4 weeks after virus injections with dental cement. After 1 week of recovery, mice were tested with a fiber photometry system (FPS-410/

470; Inper Ltd, Hangzhou, China). The excitation wavelength of 470 nm is used for calcium-dependent fluorescence emission of GCaMP6 or GCaMP7, whereas the excitation wavelength of 410 nm for calcium-independent isosbestic point (LED frequencies were 45 Hz for 410 nm and 470 nm). The final laser power was adjusted to 10−20 μW.

During the procedure, acute pruritogen His or CQ was intradermally injected into the nape of the mice, and the changes of population calcium signals were recorded aligned with the scratching behavior. The data was analyzed with Inper Data Process software (Inper Studio), and the values of calcium transients change (ΔF/F) and the area under curve (AUC) from 1 s preceding scratching onset and 5 s after scratching onset were calculated. In this experiment, ΔF is defined as the subtraction value of the recorded calcium transient and basal F value (the basal F value is defined as the average calcium transients from 1 s preceding the scratching behavior to scratching onset). Mice were perfused to examine the implant sites of the optic fibers at the end of the experiment.

### Immunofluorescent histochemical staining
Experimental procedure was performed as previous described[58]. Animals were deeply anesthetized with overdose of pentobarbital sodium (100 mg/kg, i.p.), and then perfused transcardially with 50 ml of 0.01 M phosphate-buffered saline (PBS, pH 7.4) before 200 ml of 4% paraformaldehyde in 0.1 M phosphate buffer (PB, pH 7.4). For FOS protein detection in mice with acute itch models, mice were intradermally (*i.d.*) injected with 15 μl of histamine (His; 10 μg/μl, H7250, Sigma, MO, USA) or chloroquine (CQ; 10 μg/μl, C6628, Sigma, MO, USA) dissolved with saline in the nape with saline (15 μl) injected as control. Mice were then perfused 2 h after pruritogens or saline injections with the brains removed and immersed into 30% sucrose dissolved in 0.1 M PB until they sunk to the bottom of the container. All brains were cut into 30 μm transverse sections and collected sequentially in a six-well culture plate on a freezing microtome (CM1950; Leica, Wetzlar, Germany), and prepared as free-floating sections. Sections containing injection sites or target nuclei were observed under an epifluorescence microscope (BX-60; Olympus, Tokyo, Japan) to confirm the accuracy of injections.

All serial sections prepared for immunofluorescent histochemical staining or immunohistochemical staining were blocked with 10% donkey serum for 30 min, and immersed in the mixture of primary antisera (0.02% sodium azide, 0.12% carrageenan, 0.3% Triton X-100 and 1% normal donkey serum in 0.01 M PBS) at room temperature for 16−24 h. After being rinsed in 0.01 M PBS for 3 times, sections were then incubated in secondary antisera for 4−6 h at room temperature. For DAB reaction, an ABC kit (PK-6101, Vectorlabs, CA, USA) along with DAB-nickel ammonium sulfate solution were used. All primary and second antisera utilized were listed in supplementary table 2. In some sections, the nuclear dye DAPI was used to define nuclei regions. Cell counting of FOS-expressing, eYFP-labeled and double-labeled neurons in PBN was performed manually within regions of interest on 4-5 sections from a series of every sixth section per mouse, and Image J (NIH Image) software was used in confirming the positively labeled neurons. 3 mice were used for cell counting.

### FISH staining combined with FG and FOS immunostaining
FISH staining was performed according to previous researches[59,60]. Briefly, a DIG RNA labeling kit (Roche Diagnostic, Basel, Switzerland) was used to synthesize digoxigenin (DIG)-labeled antisense single-strand RNA probes. Seven days after the injection of FG in the CM, His (15 μl, 10 μg/μl) was intradermally (*i.d.*) injected into the nape of mice to introduce an acute itch model, and mice were sacrificed 2 h afterwards. All the PB and PBS used for fixation and cryoprotection were treated with 0.1% (v/v) diethyl pyrocarbonate (DEPC; DH098-2, Genview, FL, USA) overnight, which were autoclaved ahead perfusion procedure. Sections were treated with 2% $H_2O_2$ diluted with DEPC-PB,

0.3% Triton X-100 and acetylation solution for 10 min each. After rinsing with DEPC-PB for 10 min twice, the free-floating sections were prehybridized for 1 h in a prehybridization buffer, which consisted of 5 × saline sodium citrate (SSC; 1 × SSC = 0.15 M NaCl and 0.015 M sodium citrate, pH 7.0), 2% (w/v) blocking reagent, 50% (v/v) formamide, 0.1% (w/v) N-lauroylsarcosine (NLS) and 0.1% (w/v) sodium dodecyl sulfate (SDS). VGluT2 riboprobes were then added into the prehybridization buffer with a final concentration of 1 μg/ml and reacted at 58 °C for 20 h. After rinsing with wash buffer at 58 °C for 20 min twice, sections were incubated with 20 μg/ml ribonuclease A (RNase) at 37 °C for 30 min, and subsequently rinsed in 2 × SSC and 0.2 × SSC for 20 min twice, respectively. After that, the biotinylated tyramine (BT)-glucose oxidase (GO) amplification method was used to amplify VGluT2 mRNA signals by incubating sections into a mixture (3 μg/ml GO, 1.25 μM of BT, 1% bovine serum albumin (BSA) and 2 mg/ml β-D-glucose in 0.1 M of PB) with peroxidase-conjugated anti-digoxigenin sheep antibody, rabbit anti-FG antibody and mouse anti-FOS antibody for 30 min. Sections were then incubated in FITC-Avidin, Alexa 594-donkey anti-rabbit and Alexa 647-donkey anti-mouse (supplementary table 2) for 4 h in TBS (0.1 M of Tris-HCl buffered 0.9% saline).

### Chronic itch model and erythema score
The abdominal skin and skin of nape were shaved 3 d ahead of experiment, and applied with 1-fluoro-2,4-dinitrobenzene (DNFB) that dissolved with acetone (200 μl of 0.5% DNFB) to sensitized the animal. Seven days after sensitization, the DNFB challenge (100 μl of 0.2% DNFB) were repainted on the nape of mice every other day for 2-3 weeks to evoke cutaneous reaction. Spontaneous scratching behavior was recorded for 30 min the next day of DNFB application.

In measurement of erythema score, erythema, scarring, excoriation and lichenification were used as indexes and graded on a scale of 0-3 (0: none; 1: mild; 2: moderate; 3: severe). Each individual score added up as the dermatitis score, giving the overall score scale from 0-12. The assessment was performed in a blind manner.

### Behavioral assays
Animals prepared for behavioral testing were habituated in the recording chambers for 30 min for successive 3 days. All experimental procedure and behavioral tests, including the stereotaxic injections, behavioral assays and data analysis, were performed by different testers in a blind manner.

**Acute itch model and itch behavioral test.** Mice were shaved on the nape and habituated in a cylinder chamber ($15 \times 10 \times 25$ cm$^3$) for 3 successive days ahead of behavioral assays. At the time of the behavioral tests, the spontaneous itch was regarded as the scratching behavior without pruritogen injection for 30 min. For the acute itch models, acute pruritogens His (15 μl; 10 μg/μl) or CQ (15 μl; 10 μg/μl) was intradermally (*i.d.*) injected into the midline of the nape. The scratching behavior was recorded for 30 min after injections, and the number of scratches was counted manually when either side of the hindpaw scratched the nape. In chemogenetic modulation of the CM-mPFC pathway with DREADD strategy, mice were intraperitoneally (i.p.) infused with 1 mg/kg clozapine-N-oxide (CNO; C8032, Sigma, MO, USA) dissolved in 0.9% saline 30 min ahead of pruritogens injection with saline infused as control group. In optogenetic manipulations, a consistent laser stimulation (473 nm, 20 Hz, 10 mW) was applied during the light-on phases, and the scratching behavior was recorded for 30 min in 3 min intervals in a light off-light on sequence.

**Open field test (OFT).** To evaluate the anxiety-like behavior, the OFT was operated. Mice were placed into the center of the open field arena ($50 \times 50 \times 40$ cm$^3$), and were allowed to move freely for 15 min with their movements videotaped. The apparatus was cleaned with 75%

ethanol before the following mouse was tested. Both total distance and time spent in the center of the arena were quantified using an automated infrared detection system. In pharmacogenetic manipulations, mice were placed in arena 30 min after CNO application.

**Elevated plus maze (EPM).** The EPM apparatus is consist of two opposing arms (OA, $30 \times 5$ cm), two closed arms (CA, $30 \times 5 \times 25$ cm) and a central area ($5 \times 5$ cm). Mice were placed into the central area facing one of the open arms, and were allowed to move freely for 5 min with all movements recorded. Time spent in OA (OA time) and the percentage of distance in OA (OA distance) were analyzed with Inper Data Process V.0.7.2. Likewise, in pharmacogenetic manipulations, mice were placed in arena 30 min after CNO application.

**In vitro electrophysiology**

To examine the neuronal excitability of CM in different itch models, retrograde tracer 488-retrobeads was injected into the mPFC to specifically label mPFC-projecting CM neurons. Then, to test the electrophysiological characteristics of PNs and INs in mPFC, virus labeling GABAergic interneurons in mPFC ($AAV_{2/9}$-mdlx-eGFP) was injected into bilateral mPFC. Mice from His, DNFB and the control groups were sacrificed 28 d after the virus injection. Saline injection in nape was used as the control for His group, and Acetone application as control for the DNFB group. Briefly, the brains containing target nuclei were cut into transverse slices (300 μm thick) on a vibrating microtome (Leica VT 1200 s, Heidelberger, Nussloch, Germany) in 4 °C. Slices were transferred into an incubation chamber containing artificial cerebrospinal fluid (124 mM NaCl, 2.5 mM KCl, 2 mM CaCl2, 2 mM MgSO4, 25 mM NaHCO3, 1 mM NaH2PO4, 10 mM glucose, 1 mM ascorbate and 3 mM sodium pyruvate) in 35 °C for 20 min and then in RT for no less than an hour.

The recording pipettes (3–5 MΩ) were filled with inner solution, which containing 130 mM potassium-gluconate, 5 mM NaCl, 15 mM KCl, 0.4 mM ethylene glycol tetra acetic acid (EGTA), 10 mM 4-(2-hydroxyethyl)–1-piperazineethanesulfonic acid (HEPES), 4 mM Mg-ATP, and 0.3 mM Na-GTP (adjusted to pH 7.3 with KOH). All recordings were acquired with a Digidata 1550B at 10 kHz (Axon Instruments) and low-pass filtered at 2 kHz. The recording was made with a multiclamp 700B amplifier (Axon Instruments, Foster City, CA, USA). The Clampex software (v.10.02, Axon Instruments) and Clampfit software (v.10.7, Axon Instruments) were utilized for data acquisition and analysis. Data were excluded when the resting membrane potential of neurons were more positive than −50 mV.

**Effectives of viruses.** CNO evoked currents were recorded in current-clamp mode with membrane potential be around −70 mV in chemogenetic activation procedure, and around −40 mV in inhibitory group by bath-applying CNO (10 μM) in bath. In experiment with optical stimulation (473 nm, 2 ms, 2-3 mW/mm²), whole-cell recording was performed in voltage-clamp mode, and blue light was applied through a ×40 objective lens over the field of the patched cells. The laser was given in 5 Hz, 10 Hz, and 20 Hz pulse trains to examine the effectiveness of the ChR2-expressing virus.

**Spontaneous excitatory postsynaptic currents (sEPSC).** The recordings were made in voltage-clamp (−70 mV) for 3 min, and data were analyzed using Mini Analysis software (Synaptosoft Inc.).

**Intrinsic properties.** Input resistance (Rin) was calculated with a step-wise voltage change (400 ms; −70 mV−0 mV delivered in increments of 5 mV). Rheobase was the minimal current intensity that can generate at least one action potential. Indexes were detected in current-clamped mode with supra-threshold depolarized currents of 0–200 pA (15 ms duration) that delivered in increments of 5 pA. The AP

amplitude was measured from the equipotential point of the threshold to the spike peak. Another protocol was utilized with depolarized currents of 0–200 pA (500 ms duration) in increments of 10 pA to examine the changes in the number of spikes.

**CM input activation on spiking.** Neurons were recorded in current-clamp mode, and injected with two current pulses (2 s duration; 100 pA) with a 2 s gap. A blue laser light (1 ms, 20 Hz) was delivered 200 ms before the second current pulse. The innervated spikes were recorded for 5 sweeps at an interval of 30 s.

**Monosynaptic/bisynaptic characteristics.** Tetrodotoxin (TTX; 1 μM) was utilized to block action potential-based synaptic transmission. Monosynaptic nature of synaptic connection could be proved by a rescued current after TTX and 4-aminopyridine (4-AP; 100 μM) application.

**Excitation/inhibition (E/I) ratio.** The blue laser stimulation was given through the ×40 objective lens (473 nm, 2 ms, 2-3 mW/mm²). The cell membrane potential was held at −70 mV when evoked EPSC (eEPSC) was recorded, and held at 0 mV when recording evoked inhibitory postsynaptic currents (eIPSC) with ratio calculated afterwards. The only drug used for the E/I ratio was AP-5 (50 μM). eEPSC and eIPSC onset latency was calculated as the time from stimulation onset to 10% rise time.

**Pried-pulse ratio (PPR).** The cell membrane potential was held at −70 mV, and the paired laser stimulation (2 ms; 2-3 mW/mm²) with a 50 ms interval was applied. The PPR was calculated as the ratio of peak current response to the second pulse divided by that to the first pulse.

**AMPA/NMDA ratio.** AMPA current was measured as the peak current at a holding potential of −70 mV, and NMDA current was measured 50 ms after stimulation at a holding potential of +40 mV according to previous research[61]. Picrotoxin (PTX; 10 μM) were added to the extracellular solution to block GABA receptor, and CNQX (25 μM) and AP-5 (50 μM) were added to block AMPA or NMDA current, respectively. The AMPA- and NMDA-mediated current ratio was then calculated accordingly.

During the recording process, 0.2% biocytin (Sigma-Aldrich, St. Louis, MO, USA) was added into the inner solution, and stained with 647-avidin to identify the recorded neurons.

**Cannula implantation and microinjection**

In local pharmacological modulation procedure, a bilateral cannula was implanted into the mPFC (AP: + 1.78 mm; ML: 0.00 mm; DV: − 2.00 mm) 3 weeks after the virus injections. The cannula was fixed to animals' skulls with dental cement. The microinjection apparatus consists of a Hamilton syringe (10 ml) connected to a thin poly ethylene tube. After 3 weeks of recovery, Lanicemine (10 ng/0.5 μl per site) was given within 2 min, and the inner tube of the cannula was maintained in injection sites for another 2 min to avoid spread of drug. Behavioral tests were performed 30 min after drug infusion. In DREADD modulation, CNO and Lanicemine were both applied 30 min ahead of the behavior tests. All drugs used in this study were listed in supplementary table 3.

**Statistics**

Graphpad Prism 8.4.2 software and OriginPro (v.9.8) were utilized in statistical analysis, and raw data were analyzed using paired $t$-test, unpaired $t$-test, One-way ANOVA and Two-way ANOVA for multiple comparisons. All data related were expressed as the mean ± standard error of the mean (S.E.M.), and error bars represented S.E.M. $P < 0.05$ was considered statistically significant.

**Reporting summary**

Further information on research design is available in the Nature Portfolio Reporting Summary linked to this article.

## Data availability

All data generated in this study are available upon request to the corresponding author. Source data are provided with this paper.

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

## Acknowledgements

This work was supported by STI2030-Major Projects No.2021ZD0204403, National Natural Science Foundation of China (No.82130034) and Innovation Capability Support Program of Shaanxi (No.2021TD-57) to Y.Q.L., National Natural Science Foundation of China (No. 32271045 and 31871061) and ZHUFENG Funding Program of the FMMU (No.2019rcfcdyl) to Y.L.D.

## Author contributions

The first draft of the manuscript and electrophysiological experiments were performed by J.N.L., virus injections and data collection were performed by L.J.Z., X.M.W. and H.X.S., statistical analysis was performed by J.H., F.L.W. and S.H.C., T. C., H.L., Y.L.D. and Y.Q.L. mainly contributed to the study conception and design. Y.L.D. and Y.Q.L. also revised the manuscript. All authors read and approved the final manuscript.

## Competing interests

The authors declare no competing interests.
