## [Peer Review File · Nature Communications]

Central Medial Thalamic Nucleus Dynamically Participates in Acute Itch Sensation and Chronic Itch-induced Anxiety-like Behavior in Male miceReviewers' Comments:

Reviewer #1:

Remarks to the Author:

In this manuscript, the authors aimed to identify the role of PBN-CM-mPFC pathway in itch processing. They used pharmaco-/opto-genetics, activity-dependent cell labeling, fiber photometry to investigate the role of PBN-CM and CM-mPFC pathway in itch processing. The results showed that the CM-mPFC is required for both the itch induced scratching and anxiety-like behaviors. The behavioral effect is robust, especially in the necessary experiments. Besides, they used in vitro electrophysiology to explore the excitability of CM and functional connection between the CM and mPFC. Most of electrophysiological results well supported the main conclusion of the manuscript. However, the authors should improve the readability of manuscript and provide more evidence to clarify the functional role of the CM-INmPFC and CM-PNmPFC pathway in acute and chronic itch.

Major concerns:

1. The authors used the AAV-cFos-eYFP virus to label itch-responsive PBN projecting fibers. According to the text, the development of chronic itch lasted about 2-3 weeks. The time window for c-fos driven expression of eYFP was long and there should be a lot of non-specific expression during this process, but in the control group, it doesn't look like there is much eYFP signals, which doesn't seem reasonable. The authors should provide more details about the design of this part of the experiment. Besides, Fig 2b is confusing. Which group is the graph in the upper right corner? Moreover, it will be better if the author could show the statistical results of the colocalization of eYFP+ and Fos-immunoreactive neurons.
2. The authors claimed that the PBN-CM-mPFC pathway is essential for the itch transmission and itch induced negative emotion. The results in this study only indicate that the PBN-CM pathway is required for acute itch and the CM-mPFC pathway is required for both acute itch transmission and chronic itch induced negative emotion. Authors should further examine the role of the PBN-CM pathway and the PBN-CM-mPFC pathway in chronic itch induced scratching and anxiety-like behaviors to support their conclusion.
3. Fig. 2h-2j showed that inhibition of PBN-CM pathway reduced pruritogen induced scratching behavior. The author should further examine the effect of silencing this pathway on the motor ability of mice.
4. Electrophysiological results showed a functional connection between the CM and PNmPFC in acute itch. But the functional role of the CM-PNmPFC pathway in acute itch wasn't determined. It will be nice to know whether inhibition of the CM-PNmPFC pathway could suppress acute itch induced scratching behavior.
5. Fig. 4m showed that silencing the CM-mPFC pathway in chronic itch mice increased the time spent in open arm during EPM test. Authors should further test that whether silencing the CM-mPFC pathway in naïve mice could induce similar effect during EPM test.
6. The authors claimed that CM projections in acute and chronic itch models are dynamic. It is not clear whether CM projections to INmPFC and PNmPFC are same population or not. In Fig. 6a, authors should show an example brain slice to clarify the AAV-CaMKII-ChR2-mCherry virus infected neurons in the CM.
7. In Fig.8j-l, the reviewer wonders the effect of CM-INmPFC activation on scratching and anxiety-like behaviors in chronic itch. Authors should add a set of control: Saline + Saline group. Besides, the Saline, and Saline+Lanicemine groups are better added as controls.
8. It is confusing in the Fig. 5f-o, Fig.6e-h, Fig.7g-o, Fig. 8a-c, how could the sham as a control for both His and DNFB groups when the time of treatment is far apart each other? Whether the mice were two weeks older in DNFB group than the sham/His groups? It would be better if the authors add more details, for example, when the brain slices were collected for recording? How to treat the mice in sham group? If the mice were not in the same age, it would be difficult to evaluate the comparison between DNFB and His groups. Saline group should be added as a control for His group, if the sham should be used as a control for DNFB group. Similar in behavioral tests (Supplementary Figure. 1)

Minor concerns:

1. In Fig.1h, "OPT" should be "OFT". In Fig. 4h-4k, "DIO-hM3Dq+Saline" should be "DIO-hM4Di+Saline". In addition, in page 25, line 23, "Pried-pause ratio" should be "Paired pulse ratio". In Fig.2g, it will be nice that authors can provide a better representative diagram. In the methods part, "wk" should be "weeks". In the graphical abstract, the "PrL" should be "mPFC". Authors should carefully check their data and plots. Besides, it is better to have the manuscript edited by a native speaker for better readability.
2. In Fig. 8J-I, authors showed that chronic itch induced anxiety-like behaviors was recovered in CNO + Lanicemine group. What is the effect on the chronic itch induced scratching behaviors in CNO + Lanicemine group? How were the two drugs (CNO+Lanicemine) administrated before the behavioral tests, both were given at 30 min ago?
3. The quality of images in Fig. 2c and Fig.3c should be improved. More details of projection information provided for the morphological evidence of the PBN-CM-mPFC pathway would be better.
4. In Fig.1, DNFB treatment increased scratching and anxiety-like behaviors in mice. Authors should further examine whether DNFB treatment affects the basal function of mice, such as weight and locomotor activity.
5. The GCaMP6m fluorescent signal (Figure 1a) was recorded at 470 nm or 488 nm need to be checked.
6. It's interesting that the calcium response of CM neurons decreased before scratching onset but increased after onset (Figure 1b), which is different from that of PBN-projecting CM neurons (Figure 2e). What's the relationship between the CM neurons that receive projection from PBN and the mPFC-projecting CM neurons? It would be nice if the authors could discuss this.
7. The lesion region was too local to cover the whole CM as showed in Supplementary Fig. 2, but the behaviors were significantly changed. It would be better to show the range of lesion regions (rostral-caudal distribution) and the mouse brain atlas is a good tool for precising the virus/lesion/optic fiber location information in the present study. Besides, whether the lesion in CM would affect the basic functions?
8. Calcium responses of CM were shown in only trials but not mice in Fig. 1b, c and Fig.2e, f.
9. The verification of cannula location in mPFC was missing.
10. The authors used AAV2/9-mdlx-eGFP virus to specifically label cortical GABAergic neurons. But the specificity of the virus should be verified. Authors cited Ref 15 in page 10, line 3, but this reference did not mention this virus. The authors should verify the specificity of the virus or provide reliable references which could indicate that the virus could specifically label cortical GABAergic neurons.
11. In the Fig. 5f-o, if the sham is as a control for DNFB, both functional (pre/post-) synaptic connection and intrinsic properties of mPFC-projecting CM neurons changes. It would be nice if the authors could discuss this.

Reviewer #2:

Remarks to the Author:

In their manuscript entitled "The Central Medial Thalamic Nucleus Dynamically Participates in Acute Itch Sensation and Chronic Itch-induced Negative Emotions", Li et al. dissected the functional role PBN-CM-PFC circuit in itch sensation and emotion. The PBN is critical for itch signal processing, but the downstream mechanisms remain unknown. Here the authors contribute important evidence that the central medial thalamic nucleus might be important for this process. However, there are several critical concerns.

Major points:

1. There should be images showing the virus infection area, the lesion area, the tracer injection area and so on. In some experiments, there was only a showcase image. More detailed results should be supplied.
2. The n value should be increased in some experiments. For example, the calcium imaging

experiment in Fig1 and Fig2.

3. The author used lesion manipulation in Fig1. Why not perform the chemogenetic inhibition experiment?

4. In Fig2A, the authors used cFos-EYFP. There should have baseline levels of cFos expression even in homecage mice. The author should carefully check the expression area from anterior to posterior in both DNFB and acetone groups. The FOS panel and Merged panel had different contrast in the RED channel. This kind of manipulation is not appropriate.

5. The authors showed the morphological result of the PBN-CM-PFC pathway. However, whether there are collateral projections of PBN-CM neurons? In my opinion, these morphological results were not strong enough.

6. There are several mistakes. In Fig4GK, wrong labels. In FigS5C, wrong label.

7. In Fig4, why not use optogenetic tool? Chemogenetic manipulation is not specific to manipulating projections. If there are collateral projections, this manipulation is not specific.

8. The authors showed that PNs neuronal activities were altered in Fig6FG. Whether INs neuronal activities were also changed?

9. The authors used mDLX virus. Why not use Vgat-Cre mice? At least, the authors should confirm that mDLX virus only labeled GABAergic neurons in PFC and most of GABAergic neurons were infected by this virus.

10. In Fig8, the infusion of NMDA antagonist is non-selective. Conditional knockout should be more convincing (PMID 35132099).

Minor points:

1. In Fig1B and C, why the calcium decreases before zero time point? The authors also used very small scale in heatmap bar.

2. In FigS2A, is that spot located in PVT? Or CM? And the lesion is not obvious. In S2B, the NeuN signal in the lesion group seems unchanged.

3. In FigS3 S4, the author should show the CM injection site.

4. The Fig3B showed some merged neurons in CM. However, why there were some neurons only showed EGFP.

5. Calcium imaging of CM neurons that receive PBN inputs could be added.

Reviewer #3:

Remarks to the Author:

Li et al., study transsynaptic pathway that focused on PBN-CM-PFC in mediating the sensory and affective aspects of chronic itch. They demonstrated that the CM thalamic neurons are downstream to the PBN that play a direct role in itch regulation. Authors have also shown that CM neurons send projections to mPFC and exhibit elevated transmission to mPFC GABAergic interneurons after chronic itch development. They were able to decrease chronic itch and affective behaviors by blocking excitatory transmission in the mPFC. This study is exciting and has very interesting observation, but manuscript needs some clarifications and changes before it is accepted for the publication.

Major comments:

1. Please cite primary literature, many of the citation's authors cite in the manuscript are review article. For E.g, there is quite a bit recent primary literature on circuits involved in sensory and affect processing of itch (PMIDs: 29939958, 34032210, 31554789, 29540548, 31000426, 30266741, 30819800) but authors cited a review article from 2006.

2. Supplementary figure1 shows fc-os staining for different itch stimuli and show images labelled rostral and caudal. It is not clear from the images demonstrated where CM is in the images. No anatomical landmarks available to delineate structures. Add Bregma coordinates, may be that could help the readers to see the fos distribution.

3. Authors state they use two-color fiber photometry recording system, but the signal from 40nm is just used for isosbestic point. There should be no signal related to acquire via this channel. Widely used to rule out the motion artefacts. Two-color fiber photometry reference confusing because mostly it is used to refer when using GCaMP and RCaMP. Citation for the photometry refers to multi place photometry system. Authors should cite primary papers from Rui Costa and Deisseroth labs.
4. What is the power of the laser for the photometry experiments? it is not clearly stated in the manuscript.
5. Also include information regarding where the viruses were purchased for all the experiments.
6. In Fig 2b & c, authors claim that the eyfp labelled fibers are more in chronic itch mice but it looks like the number of cells infected by the virus for the respective images are quite different. It is possible that there is variability in the injection site in terms of the labeling of the number of cell bodies. It looks like different anatomical locations were targeted in the PBN for acute and chronic itch.
7. Authors should include images for the viral injections of All the DREADD experiments, Chr2 and GtACR1 expression. It would be helpful to see the expression and spread of the virus in rostral caudal segments. These experiments are not done in any transgenic mice, so it would be good to have all these images included.
8. GtACR1 is not commonly used inhibitory opsin due to its poor trafficking and membrane targeting, it would be important to see high resolution image of its expression. Did authors see any rebound firing? How do authors know that this opsin is working. Is there any other readout other than behavior to confirm actually this opsin works?
9. In Fig 4A. retro Cre-mCh +ve cells seem to be less compared to the DIO-DREADD expressing cells. Are the DIO constructs expression independent of Cre? If that is the case then these experiments suggest that the DREADD experiments are not done in projection dependent manner.

Dear Reviewers:

Thank you for your letter and for the reviewers' comments concerning our manuscript entitled "The Central Medial Thalamic Nucleus Dynamically Participates in Acute Itch Sensation and Chronic Itch-induced Negative Emotions" (Manuscript Number: NCOMMS-22-04436A).

All comments are valuable and very helpful for revising and improving our paper, as well as all the important suggestions significant to our researches. We have studied comments carefully and have made responses point by point. Revised portions are **marked in red** in the new version. The main corrections in the paper and the responses to the reviewer's comments are as following:

Reviewer #1 (Remarks to the Author):

Major concerns:

1. The authors used the AAV-cFos-eYFP virus to label itch-responsive PBN projecting fibers. According to the text, the development of chronic itch lasted about 2-3 weeks. The time window for c-fos driven expression of eYFP was long and there should be a lot of non-specific expression during this process, but in the control group, it doesn't look like there is much eYFP signals, which doesn't seem reasonable. The authors should provide more details about the design of this part of the experiment. Besides, Fig. 2b is confusing. Which group is the graph in the upper right corner? Moreover, it will be better if the author could show the statistical results of the colocalization of eYFP+ and Fos-immunoreactive neurons.

Response: We are sorry for the lack of explanation of the experimental design in this part. The truth is, the AAV-cFos-eYFP virus was designed to be activated only by stimuli with a strong enough intensity. That is why we use chronic itch, but not acute itch model to innervate its expression. In the control group, fewer eYFP-labeled neurons could be explained by the not strong enough non-specific stimuli in inducing the eYFP expression. Additionally, the representative pictures below showed the rostral-caudal distribution of AAV-cFos-eYFP injection sites in DNFB and Acetone groups (as control). Certain number of eYFP-labeled neurons could also be observed in the control group.

The picture in the upper right corner in Fig 2b from the original version refers to FOS protein expression in DNFB group. To avoid the ambiguity, we have rearranged the representative images in both DNFB and Acetone applied groups (**Fig. 2b of new version**). Meanwhile, the statistical results of the colocalization of eYFP and FOS-immunoreactive neurons have also been added in the **Fig. 2c** in the new version (cell counting method was performed manually as described in **Page 22, Line 9-13**). The results showed that about 91.35% of the FOS-ir neurons were double-labeled with eYFP signals, suggesting the effectiveness of this virus. The percentage of eYFP/FOS double-labeled neurons in eYFP-labeled neurons was about 35.65%. This result is also understandable because virus expression is cumulative, whereas the FOS protein expression is related to transient activation of neurons.

2. The authors claimed that the PBN-CM-mPFC pathway is essential for the itch transmission and itch induced negative emotion. The results in this study only indicate that the PBN-CM pathway is required for acute itch and the CM-mPFC pathway is required for both acute itch transmission and chronic itch induced negative emotion. Authors should further examine the role of the PBN-CM pathway and the PBN-CM-mPFC pathway in chronic itch induced scratching and anxiety-like behaviors to support their conclusion.

Response: According to your valuable advice, the results of both scratching behavior and anxiety-like behavior in chronic itch model when silencing the PBN-CM pathway have both been provided

as **supplementary Figs. 3h-j**. These results indicated that no significant differences were observed in the number of scratching during light on phase in the GtACR-eGFP injected mice. Meanwhile, the pathway inhibition also showed no effect in chronic itch-induced anxiety-like behavior.

To further examine the modulatory effect of the PBN-CM-mPFC pathway, the anterograde virus AAV-hSyn-Cre was injected into the PBN, while the Cre-dependent inhibitory virus AAV-EF1 α -DIO-His-eNPHR3.0-eYFP-WPRE-hGH was injected into the CM (shown below). The optical fibers were implanted into the mPFC (AP: +1.78 mm; ML: 0.00 mm; DV: -2.20 mm) to inhibit the PBN-CM-mPFC pathway. It could be observed that in open field test, the center time increased significantly during light on phase in DIO-eNPHR3.0 injected mice, whereas no differences were observed in total distance. Meanwhile, although no significant changes could be observed in elevated plus maze test, both the OA time and OA distance exhibited an increasing trend when inhibiting the PBN-CM-mPFC pathway, which is in accordance with our primer hypothesis. The reason of this comparatively less intensified changes might due to the experimental design (potential existence of the collateral projections). Based on the insignificant changes of PBN-CM pathway in supplementary Fig. 3h-j and the results of CM-mPFC inhibition in Fig. 4, we also hypothesized that the PBN-CM-mPFC pathway might exhibit its modulatory function in anxiety-like behavior mainly through the CM-mPFC segment.

3. Fig. 2h-2j showed that inhibition of PBN-CM pathway reduced pruritogen induced scratching behavior. The author should further examine the effect of silencing this pathway on the motor ability of mice.

Response: Thank you for your valuable advice. The total distance of mice in open field test has been demonstrated in **supplementary Fig. 3i**, which indicated an unaffected motor ability of mice while silencing the PBN-CM pathway.

4. Electrophysiological results showed a functional connection between the CM and PN^{mPFC} in acute itch. But the functional role of the CM-PN^{mPFC} pathway in acute itch wasn't determined. It will be nice to know whether inhibition of the PN^{mPFC} pathway could suppress acute itch induced scratching behavior.

Response: Sorry for the lack of information in this part. To answer this question, we have specifically inhibited the CM-PN^{mPFC} pathway through DREADD strategy. The anterograde virus AAV-hSyn-Cre was injected into the CM with the AAV-CaMKII-DIO-hM4Di-mCherry injected into bilateral mPFC. The results demonstrated a significantly attenuated scratching behavior after intradermal CNO application in both His- and CQ-induced acute itch models. These results were also added into the revised manuscript (**supplementary Figs. 4b-d and in Page 8, Line 16-18**).

5. Fig. 4m showed that silencing the CM-mPFC pathway in chronic itch mice increased the time spent in open arm during EPM test. Authors should further test that whether silencing the CM-mPFC pathway in naïve mice could induce similar effect during EPM test.

Response: Thank you for your valuable advice. In fact, this experiment has also been conducted in the original manuscript, yet the data was not shown in the old version. We have added the data as

below, which indicated that no significant differences were observed in time spent in open arm while silencing the CM-mPFC pathway in naïve mice (results were also added in **supplementary Fig. 4a** and in **Page 8, Line 22-24**).

6. The authors claimed that CM projections in acute and chronic itch models are dynamic. It is not clear whether CM projections to INmPFC and PNmPFC are same population or not. In Fig. 6a, authors should show an example brain slice to clarify the AAV-CaMKII-ChR2-mCherry virus infected neurons in the CM.

Response: Since both IN^{mPFC} and PN^{mPFC} are mingled in the mPFC, it is hard to use traditional method (by injecting two different retrograde tracers into different nuclei) to examine whether there exist collateral projections from a certain subpopulation of neurons in the CM to IN^{mPFC} and PN^{mPFC}. Therefore, we injected a sparse label virus (CSSP-YFP-2E4) into the CM. There are limited number of CM neurons that were labeled with YFP, and their projecting fibers have close connections to both IN^{mPFC} and PN^{mPFC}. Although the YFP-labeled neurons in CM were not strictly divided as the same subpopulation by any chemical marker, such results indicated the connection from certain population of CM neurons to IN^{mPFC} and PN^{mPFC} simultaneously (shown as below).

An example brain slice to clarify the AAV-CaMKII-ChR2-mCherry virus infected neurons in the CM has also been added in **Fig. 6b**. We have also provided the rostral-caudal distribution of ChR2-mCherry in **supplementary Fig. 8c** to assure the accuracy of virus injections.

7. In Fig.8j-l, the reviewer wonders the effect of CM-INmPFC activation on scratching and anxiety-like behaviors in chronic itch. Authors should add a set of control: Saline + Saline group. Besides, the Saline, and Saline+Lanicemine groups are better added as controls.

Response: We are sorry for the lack of explanation of this experimental design. According to our previous results, the connections between CM and IN^{mPFC} were strengthened in the DNFB model. Since simply activating interneurons in mPFC was sufficient to induce negative emotions in mice¹, we hypothesized that the enhanced CM-IN^{mPFC} connections might partially be the explanation of anxiety-like behavior induced by the chronic itch model. Therefore, we have designed a specific activation of the CM-IN^{mPFC} pathway to **simulate** the emotional changes in chronic itch status. The results implied that CM-IN^{mPFC} pathway activation is sufficient in inducing anxiety-like behavior, suggesting the effectiveness of this simulation.

Meanwhile, according to your valuable advice, we have added the Saline+Lanicemine group as the control group for the CNO+Lanicemine group, and have added the Saline+Saline group as the control group for the Saline+Lanicemine group in **Figs. 8 j-l** of the new version. We have also added

the saline group (Saline was intradermally injected into mice) as shown below. The results indicated that no significant differences were observed in both OPT and EPM tests between Saline and Saline+Saline group. Therefore, only Saline+Saline group and Saline+Lanicemine group were added as the final version in **Figs. 8k-l**.

8. It is confusing in the Fig. 5f-o, Fig.6e-h, Fig.7g-o, Fig. 8a-c, how could the sham as a control for both His and DNFB groups when the time of treatment is far apart each other? Whether the mice were two weeks older in DNFB group than the sham/His groups? It would be better if the authors add more details, for example, when the brain slices were collected for recording? How to treat the mice in sham group? If the mice were not in the same age, it would be difficult to evaluate the comparison between DNFB and His groups. Saline group should be added as a control for His group, if the sham should be used as a control for DNFB group. Similar in behavioral tests (Supplementary Figure. 1)

Response: We are sorry for the ambiguity in this part. At first, we would like to explain the time line of this experimental design (**Fig. 7b**). According to the time line, mice from both DNFB and His groups were sacrificed around 28 d after the virus injection. The acute itch model is conducted at the same day after virus injection, only injected with His 30 min ahead of sacrifice. Twenty-eight days is also sufficient for virus expression. We have added some more details in Methods (**Page 25, Line 19-20**).

Then we would like to explain the Sham group. In the original experiment, we used Acetone as a control for the DNFB group, and the intradermal Saline injection into the nape as the control for the His group. Since there was little difference between these two groups, we pooled the data together as the original ‘Sham group’. According to your valuable advice, we added some more data in each

group, and separated these two control groups again. Therefore, the control groups in **Figs. 5-8** were shown as Saline and Acetone in the new version.

Minor concerns:

1. In Fig.1h, “OPT” should be “OFT”. In Fig. 4h-4k, “DIO-hM3Dq+Saline” should be “DIO-hM4Di+Saline”. In addition, in page 25, line 23, “Pried-pause ratio” should be “Paired pulse ratio”. In Fig.2g, it will be nice that authors can provide a better representative diagram. In the methods part, “wk” should be “weeks”. In the graphical abstract, the “PrL” should be “mPFC”. Authors should carefully check their data and plots. Besides, it is better to have the manuscript edited by a native speaker for better readability.

Response: Thank you for your detailed advice and we are sorry for our careless mistakes. The “OPT” has been changed into the “OFT” in **Fig. 1j**. The “DIO-hM3Dq+Saline” has been changed into “DIO-hM4Di+Saline” in **Figs. 4h-4k**. “Pried-pause ratio” has been changed into “Paired pulse ratio” (**Page 11, Line 21-22**). The representative diagram has also been changed into a series of schematic plot of fiber implantation in **Fig.2f**. In Methods part, all “wk” has been changed into “weeks”. The “PrL” has been changed into “mPFC” in the graphical abstract. In addition, we have also had this manuscript edited by a native speaker to improve its readability.

2. In Fig. 8J-I, authors showed that chronic itch induced anxiety-like behaviors was recovered in CNO + Lanicemine group. What is the effect on the chronic itch induced scratching behaviors in CNO + Lanicemine group? How were the two drugs (CNO+Lanicemine) administrated before the behavioral tests, both were given at 30 min ago?

Response: Thank you for your detailed review. To answer this question, we would like to explain the experimental design in detail. Our previous results have indicated a strengthened CM-IN^{mPFC} connection in DNFB model. Meanwhile, it has been proved that simply activating interneurons in mPFC was sufficient to induce negative emotions in mice¹. Therefore, we hypothesized that the chronic itch-induced anxiety-like behavior might be, at least partially, induced by the enhanced CM-IN^{mPFC} connections. Based on the above-mentioned evidence, the specific activation of the CM-IN^{mPFC} pathway was conducted to **simulate** the emotional status of mice in chronic itch model. It is why the chronic itch model was not performed in this experimental design. The results showed that

simply activating the CM-IN^{mPFC} pathway (CNO+Saline group) is sufficient to induce anxiety-like behavior, suggesting this experimental design could effectively mimic the emotional alternation as that in the DNFB mice. In addition, such behavior could be reversed by Lanicemine, the NMDA antagonist, which might indicate the participation of NMDA receptor in chronic itch-induced anxiety behavior.

Meanwhile, both CNO and Lanicemine were applied 30 min ahead of the behavior test, and detailed explanation has been added into the Methods part (**Page 28, Line 15-16**).

3. The quality of images in Fig. 2c and Fig.3c should be improved. More details of projection information provided for the morphological evidence of the PBN-CM-mPFC pathway would be better.

Response: According to your advice, more details have been added in **Fig. 2d** and images in **Fig. 3c** have been changed into more representative ones. To add more details of the PBN-CM projections to the mPFC, both low-power and high-power field images have been provided as in **Fig. 3c** in the new version.

4. In Fig.1, DNFB treatment increased scratching and anxiety-like behaviors in mice. Authors should further examine whether DNFB treatment affects the basal function of mice, such as weight and locomotor activity.

Response: Thank you for your valuable advice. The basal function of mice in the DNFB group has been added in **Supplementary Figs. 1e-g** and were also listed as below. The results showed that DNFB model has no significant impact on the basal function (including body weight and locomotor activity) of mice when compare to the baseline. We have also added the skin erythema score when conducting DNFB model in **Supplementary Fig. 1**. The related method has been added in the Methods in the new version (**Page 23, Line 23-25 and Page 24, Line 1-2**).

5. The GCaMP6m fluorescent signal (Figure 1a) was recorded at 470 nm or 488 nm need to be checked.

Response: We are sorry for this careless mistake. The fiber photometry system (FPS-410/470; Inper Ltd, Hangzhou, China) was used in our fiber photometry experiment, which used 470 nm laser light as excitation light. This mistake has been corrected in both **Fig.1** and **Fig. 2**, and in all its relevant description in Results.

6. It's interesting that the calcium response of CM neurons decreased before scratching onset but increased after onset (Figure 1b), which is different from that of PBN-projecting CM neurons (Figure 2e). What's the relationship between the CM neurons that receive projection from PBN and the mPFC-projecting CM neurons? It would be nice if the authors could discuss this.

Response: Thank you very much for your patience and detailed revisions on our manuscript. To answer this question, we have chosen two representative time point to show why there is a decrease before the onset of scratching behavior. We can see that in some cases, the scratching behavior was in the cave of the calcium signal, which might influence the final shape of the signal. Another evidence to support this hypothesis is the calcium signal of each mouse. The results from Fig. 1e showed that from the 5 mice tested, only 2 of them (mice No.1 and No.5) exhibited the decreasing trend before the onset of behavior. However, our results could not rule out the possibility of special interpretation of this cave, which requires further experiments.

7. The lesion region was too local to cover the whole CM as showed in Supplementary Fig. 2, but the behaviors were significantly changed. It would be better to show the range of lesion regions (rostral-caudal distribution) and the mouse brain atlas is a good tool for precisising the virus/lesion/optic fiber location information in the present study. Besides, whether the lesion in CM would affect the basic functions?

Response: We have changed the representative images of CM lesion in **Supplementary Figs. 2a-b**, and have presented addition rostral-caudal distribution of the lesion range with mouse brain atlas (**Supplementary Fig. 2c**). Meanwhile, according to your valuable suggestion, other basic functions like body weight and locomotor abilities were also tested in Sham, pre-lesion and post-lesion groups. The results showed that there were no significant differences in body weight and the locomotor abilities between Sham and CM lesion groups as exhibited below.

8. Calcium responses of CM were shown in only trials but not mice in Fig. 1b, c and Fig.2e, f.

Response: In this left heatmap of each Figure, each trial represents the onset of the scratching bouts in one representative mouse. Besides, we have also increased the number of mice tested, and additional heatmaps of each mouse were added in **Figs.1d,1e** and **Figs.2i, 2j**.

9. The verification of cannula location in mPFC was missing.

Response: The bilateral cannula was designed with the inner tube of the cannula 0.5 mm longer than lower edge to prevent blockage of the cannula. In this way, the injection sites would be right above bilateral mPFC. The cannula location in mPFC has been shown as below, and also in **Supplementary Fig. 8f**.

10. The authors used AAV2/9-mdlx-eGFP virus to specifically label cortical GABAergic neurons. But the specificity of the virus should be verified. Authors cited Ref 15 in page 10, line 3, but this reference did not mention this virus. The authors should verify the specificity of the virus or provide reliable references which could indicate that the virus could specifically label cortical GABAergic neurons.

Response: According to your valuable advice, we have changed the references as the original literature from Fishell's lab and another work from our lab¹, in which a similar virus from the same company with the same promotor (mdlx) was used to specifically modulate GABAergic interneurons in the mPFC.

To further specify the mdlx-eGFP expression, the immunofluorescent histochemical staining of GAD67, the marker for GABAergic neurons in the mPFC, was also performed as below. The results indicated that about 65.83% of eGFP-positive neurons were double-labeled with GAD67 in lamina II/III of the mPFC, and about 77.70% of the GAD67-labeled neurons were eGFP signal positive, indicating the effectiveness of this virus in labeling GABAergic neurons. Additionally, in electrophysiological experiment, not only eGFP signal, but also indexes like membrane capacitance (Cm) and other electrophysiological characteristics were used to confirm the recorded GABAergic interneurons (Page 10, Line 12-14).

We have also used another transgenic mouse line, the GAD2-eGFP mice, to repeat some of the key data in the electrophysiological experiments, which displayed similar results as that in the mdlx-eGFP injected mice. Combined with the above-mentioned results, we could prove the reliability of related data.

11. In the Fig. 5f-o, if the sham is as a control for DNFB, both functional (pre/post-) synaptic connection and intrinsic properties of mPFC-projecting CM neurons changes. It would be nice if the authors could discuss this.

Response: Thank you for your valuable advice. The results from Fig. 5 showed that the intrinsic properties of the mPFC-projecting CM neurons were increased in both His and DNFB groups (more activated in the DNFB group). Meanwhile, the PPR of eEPSC in PN^{mPFC} were also both decreased in His and DNFB groups, indicating the strengthened CM-PN^{mPFC} pathway in both acute and chronic itch models. However, the PPR of eIPSC in PN^{mPFC} and the PPR of eEPSC in IN^{mPFC} only decreased significantly in the DNFB model. Combined with the fact that the AMPA current of the PN^{mPFC} decreased in DNFB model, we hypothesized that the activation from CM to the IN^{mPFC} was too strong that overcame the effect of the CM-PN^{mPFC} pathway activation. We have added further discussion in the new version (**Page 17, Line 14-22**).

Reviewer #2 (Remarks to the Author):

In their manuscript entitled "The Central Medial Thalamic Nucleus Dynamically Participates in Acute Itch Sensation and Chronic Itch-induced Negative Emotions", Li et al. dissected the functional role PBN-CM-PFC circuit in itch sensation and emotion. The PBN is critical for itch signal processing, but the downstream mechanisms remain unknown. Here the authors contribute important evidence that the central medial thalamic nucleus might be important for this process. However, there are several critical concerns.

Major points:

1. There should be images showing the virus infection area, the lesion area, the tracer injection area and so on. In some experiments, there was only a showcase image. More detailed results should be supplied.

Response: Thank you for your valuable advice. We have added the schematic for the lesion site in **Supplementary Fig. 2c**, the schematic for retrograde tracer FG injection sites in **Supplementary Fig. 3c**, and the schematic for optic fiber implantation sites in fiber photometry for the PBN-CM pathway in **Fig. 2f**. We have also listed the schematic for optic fiber implantation sites in fiber photometry for the CM, the schematic for optic fiber implantation sites of GtACR-eGFP injected mice, and the cannula location in bilateral mPFC in **Supplementary Fig. 8**, and also below.

2. The n value should be increased in some experiments. For example, the calcium imaging experiment in Fig1 and Fig2.

Response: According to your suggestion, the n value of the calcium imaging experiment has been increased to 5-6 mice in both **Figs. 1a-e** and **Figs. 2e-j** in the new version. The results exhibited similar increasing trend of the calcium signal after the onset of scratching behavior in both His- and CQ- induced itch models.

3. The author used lesion manipulation in Fig1. Why not perform the chemogenetic inhibition experiment?

Response: Thank you for your valuable advice. We decided to use the lesion manipulation because the sites of lesion are comparatively more confined than that of the virus injection. CM lays right below the PVT, which exhibits complicated functions. Therefore, part of the reasons of lesion is to modulate the nucleus without affecting the surrounding nuclei.

Besides, we have also performed the chemogenetic inhibition of the CM in acute itch model in original experimental design, yet these results were not presented in the original manuscript. We have added these results as below, which demonstrated a significant relief of the scratching behavior in spontaneous and acute itch models after CNO application.

4. In Fig2A, the authors used cFos-eYFP. There should have baseline levels of cFos expression even in homecage mice. The author should carefully check the expression area from anterior to posterior in both DNFB and acetone groups. The FOS panel and Merged panel had different contrast in the RED channel. This kind of manipulation is not appropriate.

Response: Thank you very much for your patience and detailed review on our manuscript. The AAV-cFos-eYFP virus was designed to be activated only by stimuli with a strong enough intensity. There was baseline level of cFos expression in the control group, and fewer neurons were labeled with eYFP suggests that the non-specific stimuli were not strong enough to induce eYFP expression. We have also carefully checked the rostral-caudal eYFP expression (shown as below), which also demonstrated fewer eYFP-labeled neurons in PBN in the Acetone (control) group. Meanwhile, we are sorry that the inconsistent contrast of FOS panel and Merged panel. The representative images have been rearranged in **Fig. 2b** in the new version.

5. The authors showed the morphological result of the PBN-CM-PFC pathway. However, whether there are collateral projections of PBN-CM neurons? In my opinion, these morphological results were not strong enough.

Response: Thank you for your question. So far, we cannot rule out the possibility that there exist collateral projections of PBN-CM neurons from the morphological results. However, the conclusion that the PBN-CM-mPFC pathway exists is also supported by the electrophysiological technique in Fig 5 (also shown as below). After 488-retrobeads were injected into the mPFC and AAV-CaMKII-ChR2 into the PBN, the electrophysiological activity of retrobeads labeled neurons in the CM was recorded during blue light stimulation. The results that blue laser innervating ChR2-expressing fibers from the PBN could evoke eEPSC of different frequency were stronger evidence than the morphological results. From the above-mentioned results, we can, at least, ensure the existence of the PBN-CM-mPFC pathway.

6. There are several mistakes. In Fig4GK, wrong labels. In Fig55C, wrong label.

Response: We are sorry for the careless mistakes. The legend of **Figs. 4h-k** has been corrected in the new version (the “hM3Dq” has been changed into “hM4Di”). The **Supplementary Fig. 5c** has also been adjusted, and the “CamKII” has been changed into “CaMKII” in the new version.

7. In Fig4, why not use optogenetic tool? Chemogenetic manipulation is not specific to manipulating projections. If there are collateral projections, this manipulation is not specific.

Response: Thank you for your valuable advice. The chemogenetic manipulation lacks certain specificity if taking the existence of collateral projections into concern. Although we have not used optogenetic tool to innervate the CM-mPFC pathway, we have injected anterograde virus AAV-hSyn-Cre into the PBN with optogenetic inhibition virus rAAV-EF1 α -DIO-eNPHR3.0-eYFP (AAV2/9) injected into the CM. The optical fibers were implanted above the mPFC to manipulate the PBN-CM-mPFC pathway in the new version (shown as below). It could be observed that the time spent in center time increased significantly during light on phase in DIO-eNPHR3.0 injected mice, whereas no differences were observed in total distance. Meanwhile, although no significant changes could be observed in elevated plus maze test, both the OA time and OA distance exhibited an increasing trend when inhibiting the PBN-CM-mPFC pathway, which is in accordance with our primer hypothesis. The reason of this comparatively less intensified changes might due to the experimental design (potential existence of the collateral projections). Based on the insignificant changes in **supplementary Figs. 3h-j** and the results of CM-mPFC inhibition in **Fig. 4**, we also hypothesized that the PBN-CM-mPFC pathway might exhibit its modulatory functions (especially those on the anxiety-like behavior) mainly through the CM-mPFC segment.

8. The authors showed that PN neuronal activities were altered in Fig6FG. Whether IN neuronal activities were also changed?

Response: Thank you for your valuable question. We mainly presented the results of the PN^{mPFC} neuronal activities in the original version because they are responsible for the main output information from the mPFC. The data of IN^{mPFC} has also been collected in original experimental design, which is presented as below. The results showed that although the spiking number of IN^{mPFC} decreased significantly in both His and DNFB groups (a), there were no significant differences in the subtraction of No. of spikes between light-OFF and light-ON phases when summing up the spike number from 5 sweeps (b). Meanwhile, the decreasing changes of the No. of spikes during the light-OFF phases in His and DNFB groups suggest the possibility of variations in local neuronal circuit in acute and chronic itch models, which requires further experiments.

9. The authors used mDLX virus. Why not use Vgat-Cre mice? At least, the authors should confirm that mDLX virus only labeled GABAergic neurons in PFC and most of GABAergic neurons were infected by this virus.

Response: Thank you for your valuable advice. We are sorry that the transgenic mouse line were not available in the original version. However, we have managed to breed the GAD2-eGFP transgenic mice when preparing this reversion, and have repeated some of the key data in these experiments, especially the electrophysiological results. The data (as shown below and also in **Supplementary Figs. 5c-d**) displayed similar results as that in the mdlx-eGFP injected mice in Figs. 7l-o and Figs. 8a-c.

Meanwhile, we have also tested the mDLX-eGFP virus by immunofluorescent histochemical staining, which displayed that 65.83% of eGFP-positive neurons were double-labeled with GAD67 in lamina II/III of the mPFC, and 77.70% of the GAD67-labeled neurons were eGFP-positive (as shown below and also in **Supplementary Figs. 5a-b**). Although not all eGFP-labeled neurons were strictly double-labeled with GAD67, we have also used membrane capacitance (Cm) and other electrophysiological characteristics to confirm the recorded interneurons (**Page 10, Line 12-14**). Combined with the above-mentioned results, we could prove the reliability of related results.

10. In Fig8, the infusion of NMDA antagonist is non-selective. Conditional knockout should be more convincing (PMID 35132099).

Response: Thank you for your valuable advice. The transgenic mice you suggested are not for the commercial use. We have connected the original producer of the conditional knockout mice for several times, yet our e-mails turned out to be not replied. Therefore, we have tried another more selective NMDA antagonist, ketamine, to further testify the results (as shown below). Ketamine produces rapid-acting antidepressant effects in both clinical studies and in preclinical rodent models. It has been demonstrated that it is the GluN2B-NMDARs on GABAergic interneurons, but not the glutamate principal neurons that participate in its antidepressant effect². In accordance to the results in Lanicemine applied group, simply activating the CM-IN^{mPFC} pathway is sufficient to induce anxiety-like behavior (CNO+Saline group). In addition, although no significant differences could be observed between CNO+Saline and CNO+ Ketamine groups in OFT and EPM, the center time and OA time in CNO+Ketamine group showed an increasing trend, which is not different from that in the Saline+Ketamine group.

Meanwhile, we have designed another virus, AAV-CMV-DIO-(EGFP-U6)-shRNA (NR1)-WPRE-hGF (with AAV-CMV-DIO-(EGFP-U6)-shRNA(scramble)-WPRE-hGF as control), to specifically knockdown the NR1 subunit of NMDA receptors. Thanks to the GAD2-Cre transgenic mouse line, we could modulate the NMDA receptors of IN in the mPFC more specifically (**Supplementary Fig. 7**). The results indicated that NR1 knockdown of the GAD2-expressing neurons in the mPFC did not affect the Spontaneous, His- or CQ-induced scratching behavior. It, however, could significantly reduce the scratching behavior in DNFB model, as well as relieve the chronic itch-induced anxiety-like behavior. Such results are in accordance to the Lanicemine and ketamine applied groups.

Minor points:

1. In Fig1B and C, why the calcium decreases before zero time point? The authors also used very small scale in heatmap bar.

Response: Thank you very much for your patience and revisions on our manuscript. We presumed that this decrease before zero point of the previous version Fig.1b and c might stand for no specific biological meanings. Since that the $\Delta F/F$ used -1~0 s before the onset of behavior as the baseline. The decrease is more likely to be induced by the higher baseline before the behavior onset in some mice. We have also chosen two representative 470 nm signal as shown below. In some cases, the scratching behavior was in the cave of the calcium signal, which might influence the final shape of the signal. Another evidence to support this idea is the calcium signal of each mouse (**Figs. 1d-e** in revision). The results showed that from the 5 mice tested, only 2 of them (mouse No. 1 and mouse No. 5) exhibited the decreasing trend before the onset of behavior. However, our results could not rule out the possibility of special interpretation of this cave, which requires further experiments. In addition, we would like to explain that the small scale in heatmap bar in Figs. 1b-c is automatically made by Prism software according to the calcium signal from two representative mice. Based on the two representative 470 nm signal as shown below, although the scale is small, its compliance with scratching behavior is reliable.

2. In FigS2A, is that spot located in PVT? Or CM? And the lesion is not obvious. In S2B, the NeuN signal in the lesion group seems unchanged.

Response: We are sorry for the ambiguity of this lesion figure, and we have changed the representative image for a more obvious lesion spot in the CM. The NeuN-labeled neurons in original manuscript not only decreased (might not be obvious enough due to the contrast), but also exhibited a messy arrangement around the lesion sites. We presume the disordered arrangement of CM neurons also affect their original functions. However, to clarify this problem, we have changed another representative image in **Supplementary Fig. 2b** and have provided the rostral-caudal distribution of lesion sites in **Supplementary Fig. 2c**.

3. In FigS3 S4, the author should show the CM injection site.

Response: We have added the CM injection sites in **Supplementary Fig. 3b**. We have also added the injection site of AAV_{2/9}-hSyn-eGFP-Synaptophysin-mRuby virus in **Supplementary Figs. 4f-g** in the revised manuscript. The injection sites of both FG and virus were confined in CM, which ensured the reliability of the following morphological results.

4. The Fig3B showed some merged neurons in CM. However, why there were some neurons only showed EGFP.

Response: Thank you for your detailed revision on our manuscript. To answer this question, we would like to explain the mechanism of Cre enzyme in Cre/Loxp system. In fact, Cre enzyme is repeatedly used in Cre/Loxp system (a single Cre enzyme can act on multiple loxP sequences), which means that a small amount of Cre enzyme expression is sufficient for its functional purposes. It is also clear that DIO-eGFP expression could only take place with the assistance of the Cre enzyme. Therefore, it is not that there is no mCherry expression, but the amount of expression of mCherry is too weak, so that the fluorescence signal is not strong enough to be captured. We have adjusted the

lightness of the original images in **Fig. 3b**. The merged image showed that almost every eGFP-labeled neuron in the CM is double-labeled with mCherry signal.

5. Calcium imaging of CM neurons that receive PBN inputs could be added.

Response: Thank you for your valuable advice. We have added the number of mice of the PBN-CM group in the calcium imaging experiment in the new version (**Figs. 2e-l**), as well as the representative heatmap of each mouse in **Figs. 2i-j**.

Reviewer #3 (Remarks to the Author):

Li et al., study transsynaptic pathway that focused on PBN-CM-PFC in mediating the sensory and affective aspects of chronic itch. They demonstrated that the CM thalamic neurons are downstream to the PBN that play a direct role in itch regulation. Authors have also shown that CM neurons send projections to mPFC and exhibit elevated transmission to mPFC GABAergic interneurons after chronic itch development. They were able to decrease chronic itch and affective behaviors by blocking excitatory transmission in the mPFC. This study is exciting and has very interesting observation, but manuscript needs some clarifications and changes before it is accepted for the publication.

Major comments:

1. Please cite primary literature, many of the citation's authors cite in the manuscript are review article. For E.g, there is quite a bit recent primary literature on circuits involved in sensory and affect processing of itch (PMIDs: 29939958, 34032210, 31554789, 29540548, 31000426, 30266741, 30819800) but authors cited a review article from 2006.

Response: Thank you for your valuable advice. At first, we would like to explain that the citation of the review from 2006 is mainly based on its reliability, which supports the correlation between chronic itch and negative emotions. However, we do admit that some of our original citations did not keep up with the recent progress. Therefore, we have carefully read all of your recommended

works and cited some of them in our reversed manuscript (**Page 16, Line 8, reference 42-44; Page 3, Line 4, reference 2-4**).

2. Supplementary figure 1 shows cFos staining for different itch stimuli and shows images labelled rostral and caudal. It is not clear from the images demonstrated where CM is in the images. No anatomical landmarks are available to delineate structures. Adding Bregma coordinates, which may help the readers to see the Fos distribution.

Response: Thank you for your valuable advice. We have added guide lines to demonstrate CM in **Supplementary Fig. 1a**. We have also added Bregma coordinates at the lower left corner in the Saline group (Bregma -1.06 and -1.70 mm) to help the reader to see the Fos distribution.

3. Authors state they use a two-color fiber photometry recording system, but the signal from 410 nm is just used for the isosbestic point. There should be no signal related to acquire via this channel. It is widely used to rule out motion artifacts. Two-color fiber photometry is confusing because mostly it is used to refer when using GCaMP and RCaMP. Citation for the photometry refers to a multiplex photometry system. Authors should cite primary papers from Rui Costa and Deisseroth labs.

Response: We are sorry for the ambiguity and careless mistake in this part. In this experimental design, the signal from 410 nm light was used as an excitation source for the Ca²⁺-independent isosbestic control measurements, just as in Fig 1b in the original reference³. The baseline correction and motion-correction strategies were utilized in the process of data analysis by the software provided by Inper Tech in which the 410 nm signal is subtracted from the 470 nm signal. To correct this mistake, we have changed the two groups in AUC analysis (**Figs. 1f-g and Figs. 2k-l** in the reversed manuscript) into Baseline and Scratch groups. We have also added an explanation for the use of 470 and 410 nm excitation light in **Page 20, Line 23-24** and in **Page 21, Line 1**. In accordance with your advice, we have found another primary paper from Rui Costa's lab that we have cited in **Page 5, Line 4** as citation **No. 18**.

4. What is the power of the laser for the photometry experiments? It is not clearly stated in the manuscript.

Response: The power of the laser for the photometry experiments is adjusted to about 10-20 μ W. We have added this information in the new version (**Page 21, Line 1-2**).

5. Also include information regarding where the viruses were purchased for all the experiments.

Response: Thank you for your advice and we have mentioned this information in **supplementary table 1**. From this table, only AAV-CMVbGlobin-Cre-mCherry were purchased from the Shanghai Taitool Bioscience Co. Ltd. Other viruses were all purchased from the Wuhan Brain VTA Co. Ltd.

6. In Fig 2b & c, authors claim that the eyfp labelled fibers are more in chronic itch mice but it looks like the number of cells infected by the virus for the respective images are quite different. It is possible that there is variability in the injection site in terms of the labeling of the number of cell bodies. It looks like different anatomical locations were targeted in the PBN for acute and chronic itch.

Response: We are sorry for the ambiguity in this part. The fact is, both mice from the Acetone or the DNFB group were injected with the same virus according to the same PBN coordinates. In this experiment, the AAV-cFos-eYFP virus (activity-dependent virus) was used. This virus fused with c-fos as a promoter was designed to be activated only by stimuli with certain intensity. Since that chronic itch has been proved to induce FOS expression, so we can observe a large number of neurons in the PBN and abundant axonal terminals projecting from PBN to the CM of in mice from DNFB model. However, the Acetone group is the control group, and the mice did not experience such strong stimulus as that in chronic itch group. Therefore, it is understandable that the eYFP-labeled neurons in Acetone group were comparatively less than that in the DNFB applied group.

To further clarify this problem, we have also provided a rostral-caudal distribution of virus expression as shown below. The results showed that there are more eYFP-labeled neurons in the DNFB group than that in the control group in slices from any Bregma coordinate, indicating the effectiveness of this activity-dependent virus.

7. Authors should include images for the viral injections of All the DREADD experiments, Chr2 and GtACR1 expression. It would be helpful to see the expression and spread of the virus in rostral caudal segments. These experiments are not done in any transgenic mice, so it would be good to have all these images included.

Response: Thank you for your advice. We have included the rostral-caudal distribution of the injection sites in DREADD experiments, Chr2-mCherry injected groups, as well as the schematic for optic fiber implantation sites of GtACR-eGFP injected mice in **Supplementary Fig. 8**.

8. GtACR1 is not commonly used inhibitory opsin due to its poor trafficking and membrane targeting, it would be important to see high resolution image of its expression. Did authors see any rebound firing? How do authors know that this opsin is working. Is there any other readout other than behavior to confirm actually this opsin works?

Response: Thank you for your valuable suggestion. We have added the high-resolution image of GtACR expression as below. To further clarify the effectiveness of this opsin, we have conducted an electrophysiological experiment. GtACR-eGFP labeled neurons were recorded in the current

mode (clamped at -50 mV), and were performed with a light off-light on-light off three-phases protocol (15 s for each phase; 1-ms pause, 20 Hz, 2-3 mW/mm²). The results showed that blue laser light could effectively reduce the spontaneous firing of these CM neurons, which indicated the effectiveness of this virus.

9. In Fig 4A. retro Cre-mCh +ve cells seem to be less compared to the DIO-DREADD expressing cells. Are the DIO constructs expression independent of Cre? If that is the case then these experiments suggest that the DREADD experiments are not done in projection dependent manner.

Response: We are sorry for the lack of explanation in this part. To answer this question, we would like to clarify the mechanism of Cre enzyme. It is repeatedly used in the Cre/Loxp system (a single Cre enzyme can act on multiple loxP sequences), which means that a small amount of Cre expression is sufficient for its functional purposes. It is also clear that DIO-DREADD expression could only take place with the assistance of the Cre enzyme. Therefore, it is not that there is no expression, but that the amount of expression of mCherry is too weak, so that the fluorescence signal is not strong enough to be captured. To further support this idea, we have provided the same representative picture with higher light intensity as shown below. The results indicated almost every eGFP-labeled neuron is double-labeled with mCherry signal (white arrow heads).

Other changes:

We have tried our best to improve the manuscript according to your comments and valuable suggestions. Some other small changes have also been made and marked in red in the manuscript, which will not influence the content or framework of this work.

We have also arranged an Excel file containing all the original data set from all figures, which could be provided once upon required. We appreciate for Editors/Reviewers' warm work earnestly, and hope that the correction will meet with approval. Once again, thank you very much for your comments and suggestions.

References

- [1] Yin, J. B., Liang, S. H., Li, F., Zhao, W. J., Bai, Y. & Sun, Y. et al. dmPFC-vlPAG projection neurons contribute to pain threshold maintenance and antianxiety behaviors. *J. Clin. Invest.* **130**, 6555-6570 (2020).
- [2] Gerhard, D. M., Pothula, S., Liu, R., Wu, M., Li, X. & Girgenti, M. J. et al. GABA interneurons are the cellular trigger for ketamine's rapid antidepressant actions. *J. Clin. Invest.* **130**, 1336-1349 (2020).
- [3] Kim, C. K., Yang, S. J., Pichamoorthy, N., Young, N. P., Kauvar, I. & Jennings, J. H. et al. Simultaneous fast measurement of circuit dynamics at multiple sites across the mammalian brain. *Nat. Methods.* **13**, 325-328 (2016).

Reviewers' Comments:

Reviewer #1:

Remarks to the Author:

The authors answered most of the questions in detail, but there are some minor issues.

1. In Fig. 2c, authors should show the percentage of the eYFP/Fos double-labeled neurons that overlap with Fos- or eYFP-expressing neurons in the PBN in Acetone group as well.
2. In response letter, authors showed that inhibition of PBN-CM-mPFC pathway decreased scratching behaviors in acute itch model but increased anxiety-like behaviors in chronic itch model. This is unexpected. Authors should discuss it.
3. In supplementary Fig. 3, authors showed that inhibition of PBN-CM pathway has no effect on chronic itch. In addition, in response letter, authors showed that inhibition of PBN-CM-mPFC pathway increased anxiety-like behaviors in chronic itch model. This is also unexpected, because the CM neurons that receive the PBN input and project to mPFC should also be included when manipulating the PBN-CM circuit. Authors should discuss this result.
4. In legend of Supplementary figs. 3, "OPT: open filed test;" should be "OFT: open field test".

Reviewer #2:

Remarks to the Author:

The authors have revised the draft and there was a great improvement. However, there are still several concerns.

1. In page 4 line 19, "The DNFB model has no effect on the basal function of mice". The description is too subjective.
2. In calcium experiments, what's the meaning of $\Delta F/F$? There was no description for basal F. Was there a basal time window (such as -1 to 0) for normalising the calcium signal in each trial?
3. In page 5 line, the authors compared results of three groups, the ace+sham, DNFB+sham and DNFB+lesion. There should be a ace+lesion group. Lesion PVT might cause robust behavioral changes.
4. In Fig 2d, the figs still have different contrast issue.
5. In Fig. 2b, same bregma brain slices should be provided. Obviously, the pbn in ace group was posterior than the pbn in dnfb group.
6. In page 6, the authors could calculate the percentage of FG+Fos+/FG+.
7. In fig 2g-j, the showcase in g and h used a relatively small hotmap (-0.2 to 0.8), and the summary in i and j used -5 to 30 hotmap. What's the reason?
8. In page 7 sfig 3j, half of the gfp control mice did not enter the OA zone and this caused the relatively short duration and less distance in control group. Using this kind of behavioral data was not recommended.
9. In page 7, line 13, 'CM pathway is necessary for itch sensation' should be revised. The data showed the itch scratching behaviors were affected. Whether the sensation or motivation was affected need more experiments.
10. In fig 3b,e, the authors should show the CM boundary clearly. The current images were not appropriate as readers could not know whether the virus infected nuclei around CM. Same problem appeared in the behavioral experiments. The authors should provide the verification image of each mouse used in the experiments to convince the readers. If the infection was beyond the CM, other pathway was involved and could not draw the conclusion.
11. In page 8, line 14, whether there are collateral projections from CM-mPFC pathway? The current results only suggested silencing mPFC-projecting neurons in cm affected scratching behaviors.
12. The PBN-CM seems to be involved in both pain and itch signals processing. Whether the same population of pbn neurons participate in these modalities?

Reviewer #3:

Remarks to the Author:

I am satisfied with the changes made to the manuscript. I recommend the manuscript for publication with minor revision. I would like the authors to provide more information regarding the LED frequencies they have used to trigger 410 and 470nm LEDs in photometry experiments.

Dear Reviewers:

Thank you again for your letter and for the reviewers' comments concerning our manuscript entitled "The Central Medial Thalamic Nucleus Dynamically Participates in Acute Itch Sensation and Chronic Itch-induced Negative Emotions" (Manuscript Number: NCOMMS-22-04436B).

All comments are valuable and very helpful for revising and improving our paper, as well as all the important suggestions significant to our researches. We have studied all these comments carefully and have made responses point by point. All revised context is marked in red in the new version. The main corrections in the paper and the responses to the reviewers' comments are as following:

Reviewer #1 (Remarks to the Author):

The authors answered most of the questions in detail, but there are some minor issues.

1. In Fig. 2c, authors should show the percentage of the eYFP/Fos double-labeled neurons that overlap with Fos- or eYFP-expressing neurons in the PBN in Acetone group as well.

Response: We are sorry for the missing of this information in Fig. 2c, and have provided addition information about this in the Acetone group in new version. It worth noticing that the percentages of eYFP/Fos double-labeled neurons that overlap with Fos- or eYFP-expressing neurons did not exhibit much difference between the DNFB and Acetone groups, which is understandable because of the intrinsic properties of this virus (The viral expression can be activated in the presence of strong stimuli). However, the absolute value of the eYFP-labeled neurons in DNFB group is significantly higher than that in the Acetone group (exhibited as below). Such results are in accordance with our hypothesis that chronic itch stimuli could induce PBN activation in DNFB model.

2. In response letter, authors showed that inhibition of PBN-CM-mPFC pathway decreased

scratching behaviors in acute itch model but increased anxiety-like behaviors in chronic itch model. This is unexpected. Authors should discuss it.

Response: Thank you for your detailed review. We would like to add further discussion to explain these results. The main idea of this research is to **distinguish the distinct status of the PBN-CM-mPFC pathway in acute and chronic itch models. It implies that the same pathway may play a different role in different itch models** because of the strength of CM projections changes when connect to PNs or INs in the mPFC.

In acute itch model, the CM has strengthened connections with PNs and mainly participates in acute itch sensation. Therefore, inhibition of this pathway could attenuate scratching behavior. However, the CM connections to INs were enhanced with chronic itch stimuli, and the enhanced CM-mPFC pathway is proved to modulate chronic itch induced anxiety-like behavior. In this case, it is understandable that inhibition of this pathway in chronic itch model exhibited a **relief** of anxiety-like behavior, although this trend could only be observed in open field test (**the center time in open field test increased in DNFB model while inhibiting the PBN-CM-mPFC pathway**).

3. In supplementary Fig. 3, authors showed that inhibition of PBN-CM pathway has no effect on chronic itch. In addition, in response letter, authors showed that inhibition of PBN-CM-mPFC pathway increased anxiety-like behaviors in chronic itch model. This is also unexpected, because the CM neurons that receive the PBN input and project to mPFC should also be included when manipulating the PBN-CM circuit. Authors should discuss this result.

Response: Thank you for your detailed review and indeed, this question should not be ignored. At first, we would like to mention that inhibiting the PBN-CM-mPFC pathway **relief** the anxiety-like behaviors induced by chronic itch model, since that the center time increased in open field test. Then, we would like to explain this scenario by emphasizing that the modulatory effect of PBN-CM pathway and the PBN-CM-mPFC pathway are **not necessarily the same**.

Except from the mPFC, CM also send abundant projections to a series of nuclei in the forebrain, like the insular cortex, the amygdala and nucleus accumbens¹. Each of them might play essential roles in itch sensation². Therefore, it is understandable that the modulatory effect of the PBN-CM pathway and the PBN-CM-mPFC pathway may distinct from each other. Such results implied that the modulatory role of the PBN-CM-mPFC pathway in chronic itch model could be, at least partially, explained by the mPFC-projecting neurons in the CM (**pathway-specific modulatory effect**).

4. In legend of Supplementary figs. 3, “OPT: open filed test;” should be “OFT: open field test”.

Response: We are sorry for this careless mistake, and have changed “OPT” into the “OFT” in the legend of Supplementary fig. 3.

Once again, thank you for your detailed review and all of your valuable suggestions.

Reviewer #2 (Remarks to the Author):

The authors have revised the draft and there was a great improvement. However, there are still several concerns.

1. In page 4 line 19, "The DNFB model has no effect on the basal function of mice". The description is too subjective.

Response: We are sorry for the too subjective remarks, and have specified the “basal function” into the items that we have examined, like the body weight and locomotor abilities. The original sentence has been changed into “The DNFB model has no effect on the body weight or the locomotor abilities of mice” in **Page 4, Line 19-20** in the new version.

2. In calcium experiments, what's the meaning of delta F/F? There was no description for basal F. Was there a basal time window (such as -1 to 0) for normalising the calcium signal in each trial?

Response: Thank you for your valuable suggestion, and we are sorry that we have missed the detailed description for the basal F in Method. The average value of calcium transients in -1 to 0 s (basal time window) is used as the baseline to normalize the calcium transients in each trial, and the ΔF value is the subtraction of recorded value and basal F value. Such descriptions have been added and marked in red in **Page 21, Line 10-12** in the new version.

3. In page 5 line, the authors compared results of three groups, the ace+sham, DNFB+sham and DNFB+lesion. There should be an ace+lesion group. Lesion PVT might cause robust behavioral changes.

Response: Thank you for your valuable suggestions. To address this problem, we have added the ace+lesion group in **Figs. 11 and 1m** in the new version. We have also added these results here in the response as below. The results showed that no significant differences could be observed between Actetone+Sham and Acetone+Lesion groups, although the OA time and OA distance in Acetone+Lesion group exhibited a decreasing trend. The results indicated that lesion CM might induce minor changes in anxiety-like behavior, yet not robust enough to make a significant difference.

4. In Fig 2d, the figs still have different contrast issue.

Response: We are sorry for the careless mistake, and we have adjusted the contrast of two images in Fig 2d as the same in the new version.

5. In Fig. 2b, same bregma brain slices should be provided. Obviously, the pbn in ace group was posterior than the pbn in dnfb group.

Response: Thank you so much for your detailed review. We have adjusted the PBN image in

Acetone group into a new set, which is closer with that in DNFB group in bregma. The results showed similar trend as that in original version, and the eYFP-labeled neurons is fewer than that in DNFB group. Meanwhile, such results are in accordance with our hypothesis that chronic itch stimuli could induce PBN activation in DNFB model.

6. In page 6, the authors could calculate the percentage of FG+Fos+/FG+.

Response: Thank you for your valuable advice, and we have added the percentage of FG+FOS+/FG+ in both **Fig. S3 e** and in page 6 (**Page 6, Line 13**; also exhibited as below). The results showed that about 64.35% of the FG-labeled neurons were activated by acute itch stimuli.

7. In fig2g-j, the showcase in g and h used a relatively small hotmap (-0.2 to 0.8), and the summary in i and j used -5 to 30 hotmap. What's the reason?

Response: We are sorry for the careless mistake. The showcase in g and h are automatically generated by analysis software, and the -0.2 to 0.8 should be shown in percentages as -20% to 80%.

We have changed the showcase in Fig 2g-j in the new version.

8. In page 7, Fig. 3j, half of the GFP control mice did not enter the OA zone and this caused the relatively short duration and less distance in control group. Using this kind of behavioral data was not recommended.

Response: We are sorry for the lack of explanation in Fig. S3j. In this experiment, the mice were performed with **DNFB model**, and the AAV-hSyn-DIO-eGFP injected group (into DNFB mice) was used as the control group. Therefore, it is understandable that mice in control group exhibited relatively shorter OA time and smaller percentages of OA distance (%), which indicated that the DNFB model was successfully conducted. To prevent future misunderstanding in these results, we have added “DNFB model” written in **Fig. S3f**, and in figure legend of Fig. S3 in the new version.

9. In page 7, line 13, 'CM pathway is necessary for itch sensation' should be revised. The data showed the itch scratching behaviors were affected. Whether the sensation or motivation was affected need more experiments.

Response: Thank you for your valuable advice, and we do admit this description is too subjective based on our current results. We have changed this comment into “the PBN-CM pathway is necessary for itch-induced scratching behavior” in **Page 7, Line 14** in the new version.

10. In Fig. 3b,e, the authors should show the CM boundary clearly. The current images were not appropriate as readers could not know whether the virus infected nuclei around CM. Same problem appeared in the behavioral experiments. The authors should provide the verification image of each mice used in the experiments to convince the readers. If the infection was beyond the CM, other pathway was involved and could not draw the conclusion.

Response: Thank you so much for your advice, we do admit that the accuracy of injection sites is important in this experiment. Therefore, we have added the CM boundary in **Figs 3b and 3f** in the new version, and also as below. The figure legend has also been adjusted accordingly.

Secondly, the accuracy of injections strongly affects the reliability of the behavioral results. Although the injection sites of each mouse had been verified after behavior tests, we are sorry that not all of them have been photographed. Meanwhile, due to the large number of mice used in this experiment, as well as the limited space in all Figures, it is difficult to produce the injection sites of ALL mice in this experiment. However, we did confirm the injection sites in each mouse, whose behavior data were included in analysis, and some of the injection sites were listed as below. These evidence, although not for each mouse, all proved that our injections are accurate enough for further behavior experiments.

The CM is located ventrally of paraventricular nucleus of the thalamus (PVT). So the next, we would like to explain the relationship of PVT and CM, since that PVT is the nucleus that was most likely to be affected by a small amount of spread of viruses in some pictures we have provided (yet all very limited). According to our other research (not published yet), activating PVT using DREADD technology significantly **alleviated** scratching behavior (i.e. PVT has opposite effect to the CM in itch-induced scratching behavior). Therefore, even if there is little leakage of virus infection into the PVT in some occasions, the behavior results of CM modulation are convincing enough. All evidence above has proved the reliability of our behavior results.

a Injection sites of eGFP-labeled viruses into CM with nuclear dye DAPI

b Injection sites of eGFP-labeled viruses into CM

c Injection sites of mCherry-labeled viruses into CM

11. in page 8, line 14, whether there is collateral projections from CM-mPFC pathway? The current results only suggested silencing mPFC-projecting neurons in cm affected scratching behaviors.

Response: Thank you for your detailed review, and we have designed another viral strategy to address this problem. To avoid the possible collateral projections from the mPFC-projecting neurons in the CM, we have injected anterograde virus AAV (2/1)-hSyn-Cre-mCherry into the CM, along with Cre-dependent AAV (2/9)-hSyn-DIO-hM4Di-eGFP into bilateral mPFC. This experimental design allows specific inhibition of the CM-mPFC pathway. The results showed that specific CM-mPFC pathway inhibition using anterograde viral strategy significantly suppressed His-, CQ-induced acute itch, as well as attenuated the scratching behavior and the anxiety-like behavior induced by chronic itch model. Such results exhibited similar trend as that in the original retrograde viral strategy, which implies that CM-mPFC pathway is strong enough to modulate scratching

behavior although the collateral projection from the mPFC-projecting neurons in CM might exist.

12. The PBN-CM seems to be involved in both pain and itch signals processing. Whether the same population of pbn neurons participate in these modalities?

Response: So far, the molecular defined itch pathway and known pain circuits have been proved to overlap both anatomically and functionally, and PBN is one of the key nuclei in this process³. It has been proved that VGLUT2⁺ neurons in PBN modulate itch sensation, and conditional knockout of VGLUT2⁺ neurons in PBN significantly alleviate scratching behavior⁴. PBN also plays essential roles in nociception, since that Tacr1⁺ neurons in superior lateral PBN participate in pain signal transmission to intralaminar thalamic nuclei^{5,6}. Although there is still little evidence in clarifying the exact relationship of a certain subpopulation of PBN neurons in both pain and itch sensation, one research has proved that activating Tacr1⁺ neurons in PBN dramatically strengthened nociceptive behaviors and suppress itch⁷. Meanwhile, VGLUT2⁺ neurons are the most abundant neurons in PBN⁴, and the distribution area of VGLUT2⁺ and Tacr1⁺ are overlapped with each other⁵. Therefore, although evidence remained scarce, there are possibilities that certain subpopulation of neurons in PBN participate in both pain and itch sensation.

Once again, thank you for your detailed review and all your valuable suggestions.

Reviewer #3 (Remarks to the Author):

I am satisfied with the changes made to the manuscript. I recommend the manuscript for publication

with minor revision. I would like the authors to provide more information regarding the **LED frequencies** they have used to trigger 410 and 470nm LEDs in photometry experiments.

Response: We have added the LED frequencies (45 Hz for both 410 nm and 470 nm) in Methods (Page 21, Line 2-3) . Thank you again for your detailed review and all your valuable suggestions.

References

1. Vertes, R. P., Linley, S. B. & Hoover, W. B. Limbic circuitry of the midline thalamus. *Neuroscience & Biobehavioral Reviews*. **54**, 89-107 (2015).
2. Najafi, P. et al. Central mechanisms of itch: A systematic literature review and meta-analysis. *J. Neuroradiol.* **47**, 450-457 (2020).
3. Piyush, S. D. & Barik, A. The Spino-Parabrachial Pathway for Itch. *Front. Neural Circuits*. **16**, 805831 (2022).
4. Mu, D. et al. A central neural circuit for itch sensation. *Science*. **357**, 695-699 (2017).
5. Deng, J. et al. The Parabrachial Nucleus Directly Channels Spinal Nociceptive Signals to the Intralaminar Thalamic Nuclei, but Not the Amygdala. *Neuron*. **107**, 909-923 (2020).
6. Choi, S. et al. Parallel ascending spinal pathways for affective touch and pain. *Nature*, (2020).
7. Barik, A. et al. A spinoparabrachial circuit defined by Tacr1 expression drives pain. *Elife*. **10**, (2021).

Reviewers' Comments:

Reviewer #1:

Remarks to the Author:

The authors have addressed all the concerns. No more comments.

Reviewer #2:

Remarks to the Author:

I am satisfied with the changes made to the manuscript. I only have some advice for the discussion section. There are several studies focused on dissecting the functional roles of PBN downstream pathways (PMID: 32289251,35167440). These related works help to understand the precise circuit mechanisms of PBN.

Dear Reviewers:

Thank you again for your letter and for the reviewers' comments concerning our manuscript entitled "The Central Medial Thalamic Nucleus Dynamically Participates in Acute Itch Sensation and Chronic Itch-induced Negative Emotions" (Manuscript Number: NCOMMS-22-04436C).

We would like to address the final concern from reviewer 2:

I am satisfied with the changes made to the manuscript. I only have some advice for the discussion section. There are several studies focused on dissecting the functional roles of PBN downstream pathways (PMID: 32289251,35167440). These related works help to understand the precise circuit mechanisms of PBN.

Response: Thank you for your valuable advice. We have carefully read two papers you mentioned, and we admit that the subpopulation of PBN neurons (divided by different efferent nuclei) significantly influence their effects in nociception. Meanwhile, the PBN-PVT pathway also plays modulatory roles in negative affective states. All these researches emphasize the importance of PBN precise circuit mechanism, which is less mentioned in the present manuscript, and could be the focus of our future studies. We have also added the reference in Page 15, line 13. Once again, thank you for your detailed review and your valuable suggestions.